# Privacy-Preserving and Effective Cross-City Traffic Knowledge Transfer via Federated Learning

## Abstract

Traffic prediction aims to forecast future traffic conditions using historical traffic data, serving a crucial role in urban computing and transportation management. While transfer learning and federated learning have been employed to address the scarcity of traffic data by transferring traffic knowledge from data-rich to data-scarce cities without traffic data exchange, existing approaches in Federated Traffic Knowledge Transfer (FTT) still face several critical challenges such as potential privacy leakage, cross-city data distribution discrepancies, and low data quality, hindering their practical application in real-world scenarios. To this end, we present FedTT, a novel privacy-aware and efficient federated learning framework for cross-city traffic knowledge transfer. Specifically, our proposed framework includes three key innovations: (i) a traffic view imputation method for missing traffic data completion to enhance data quality, (ii) a traffic domain adapter for uniform traffic data transformation to address data distribution discrepancies, and (iii) a traffic secret aggregation protocol for secure traffic data aggregation to safeguard data privacy. Extensive experiments on 4 real-world datasets demonstrate that the proposed FedTT framework outperforms the 14 state-of-the-art baselines. All code and data are available at https://anonymous.4open.science/r/FedTT.

## 1  Introduction

**Traffic Prediction** (TP) [70, 51, 80] leverages widespread sensors in the road network to forecast traffic conditions based on historical traffic data (e.g. traffic flow, speed, and occupancy), which not only facilitates the effective allocation of public transportation resources [45] but also contributes to alleviating traffic congestion [74]. To achieve accurate TP, numerous methods have been proposed [80, 23, 24], which typically rely on a large number of traffic data to train high-performing traffic models. However, urban traffic data is often insufficient or unavailable [36, 63, 65], particularly in emerging cities, such as developing regions in the Midwestern United States [1], where sensors are newly deployed or data collection is still in its early stages. In such cases, training traffic models becomes particularly challenging and prone to overfitting, limiting the accuracy of TP tasks [27, 46].

**Transfer Learning** (TL) [59, 13], a knowledge transfer paradigm, has been widely adopted in TP scenarios to address the scarcity of traffic data. To improve the performance of traffic models in data-scarce target cities, existing TL-based TP methods [41, 43, 57] transfer traffic knowledge from data-rich source cities to target cities, which typically rely on centralized frameworks and involve the exchange of traffic data among cities

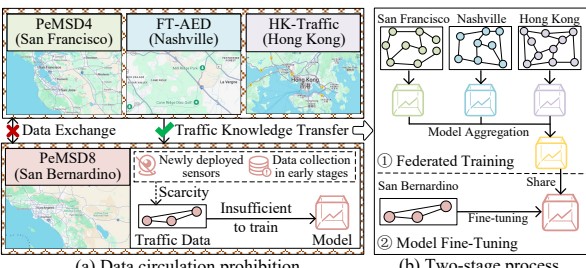

(a) Data circulation prohibition  (b) Two-stage process

Figure 1: Privacy-preserving traffic knowledge transfer

without considering data privacy. However, the direct sharing of traffic data risks privacy leakage [39, 45, 70] as such data may contain sensitive personal information. For example, sparse traffic flow data may allow attackers to infer the presence and approximate locations of individual vehicles [6, 7]. Besides, many privacy laws and regulations, such as GDPR [5] and CCPA [4], mandate data collectors to minimize non-essential data transmission and avoid centralized data storage. Therefore, maintaining the decentralization of traffic data in TP is critical. As shown in Fig. 1(a), PeMSD4 [3], FT-AED [12], HK-Traffic [2], and PeMSD8 [3] are four real-world traffic datasets, which correspond to the cities of San Francisco (**SF**), Nashville (**NV**), Hong Kong (**HK**), and San Bernardino (**SB**), respectively. Among these, SF, NV, and HK represent source cities, while SB serves as the target city. Due to legal restrictions, traffic data cannot be exchanged among cities, meaning each city can only access its local data. In this case, transferring traffic knowledge from these three source cities to the target city without exchanging raw traffic data becomes challenging.

**Federated Learning** (FL) [68, 37, 70], a privacy-preserving distributed learning paradigm, has been widely used in numerous applications to address privacy concerns such as urban computing [66] and transportation management [70]. For instance, JD Company (one of the largest e-commerce companies in China) developed the Fedlearn platform to help protect data privacy for TP applications [20]. Inspired by its success, recent studies [49, 78] have explored the FL framework to transfer traffic knowledge while preserving data privacy, which typically follow a two-stage process, as illustrated in Fig. 1(b). In the first stage, the three source cities (i.e., SF, NV, and HK), as clients, use their local traffic data to train individual local models. Subsequently, clients upload training gradients or model parameters to a central server, which aggregates to a global traffic model and then broadcasts the global model back to clients for local model updates. This process iterates until the global model converges. In the second stage, the converged global model is shared with the target city (i.e., SB) and further fine-tuned using its local traffic data. While this two-stage knowledge transfer framework has become the mainstream approach in **Federated Traffic Knowledge Transfer** (FTT), it faces three unresolved challenges, i.e., privacy, effectiveness, and robustness, that hinder its application in real-world traffic knowledge transfer scenarios, as illustrated in Fig. 2.

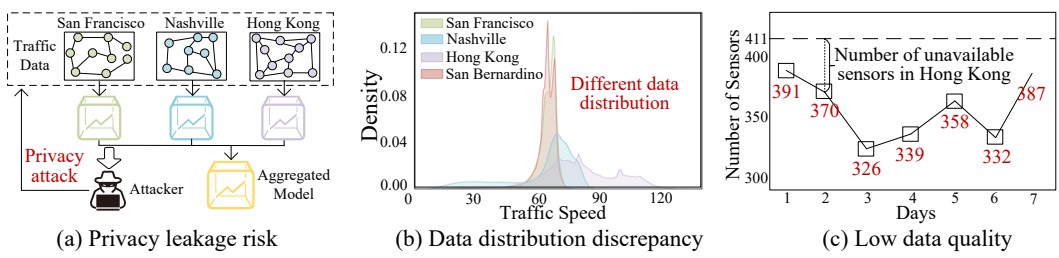

(a) Privacy leakage risk      (b) Data distribution discrepancy      (c) Low data quality

Figure 2: Four unresolved challenges in federated traffic knowledge transfer (FTT)

*Challenge 1: How to effectively protect data privacy in FTT?* Although existing methods utilize FL to avoid raw data exchange, there remains a potential risk of data privacy leakage. This arises because these methods require the uploading of training gradients or model parameters for aggregation in FTT, which may allow attackers to infer raw data by inference attacks [18, 67, 81], as depicted in Fig. 2(a). To mitigate this risk, a straightforward approach is to apply privacy-preserving techniques such as Homomorphic Encryption (HE) [52] and Differential Privacy (DP) [16] for secure aggregation on the uploaded data. However, HE introduces significant computation and communication overheads, which diminishes training efficiency, while DP lowers data utility and thus decreases model accuracy, as proved by previous studies [64, 58, 15]. Therefore, how to effectively safeguard data privacy in FTT without compromising training efficiency and model accuracy remains a significant challenge.

*Challenge 2: How to mitigate the impact of cross-city data distribution discrepancies on FTT?* None of the previous studies have considered the discrepancies in traffic data distribution across cities, which decreases the effectiveness of traffic knowledge transfer [41, 43, 57]. Specifically, the traffic domain varies significantly across cities, with distinct distributions of traffic flow, speed, and occupancy data. As shown in Fig. 2(b), we illustrate the frequency density distribution of traffic speed data for SF, NV, HK, and SB. As observed, SF and SB exhibit similar data distributions, suggesting closely related traffic domains, while NV and SB show different data distributions, indicating quite distinct traffic domains. Consequently, traffic knowledge transfer from SF to SB results in smaller prediction errors and is more effective than the transfer from NV to SB. Overall, how to address traffic domain discrepancies across cities to improve the effectiveness of FTT is an urgent challenge.

*Challenge 3: How to overcome low traffic data quality issues in FTT?* Existing methods assume that traffic data is consistently high-quality and reliable, neglecting the prevalence of missing data. As shown in Fig. 2(c), we illustrate the number of available sensors over a week in HK, which has 411 sensors in total. Due to sensor failures or updates [73, 50], the number of available sensors in HK may fluctuate over time, disrupting the model training process. While some data imputation methods [8, 48, 73] can be employed to complete missing data, they fail to effectively capture the spatio-temporal dependencies inherent in traffic data, leading to suboptimal accuracy. Consequently, how to enhance the traffic data quality to improve the robustness of FTT is another challenge.

**Contributions.** To address these challenges, we propose FedTT, a privacy-preserving and efficient F̲ederated learning framework for cross-city T̲raffic knowledge T̲ransfer. Unlike existing FTT methods, FedTT transforms the traffic data from the source cities' domain to the target city's domain and training the target city's model on the transformed data. To address *Challenge 1*, FedTT introduces the Traffic Secret Aggregation (TSA) protocol to securely aggregate the transformed data without compromising training efficiency or model accuracy. To overcome *Challenge 2*, FedTT develops the Traffic Domain Adapter (TDA) to uniformly transform the traffic data from source cities' domains to that of the target city through traffic domain transformation, alignment, and classification. To deal with *Challenge 3*, FedTT designs the Traffic View Imputation (TVI) method to complete missing traffic data by capturing the spatio-temporal dependencies. Finally, extensive experiments conducted on 4 real-world datasets demonstrate that FedTT achieves state-of-the-art performance, reducing prediction MAE by **5.43% to 75.24%** and maintaining Pearson Correlation Coefficient (PCC) of data reconstruction attacks at **no more than 10%** compared to 14 baseline methods.

## 2 Problem Definitions

The frequently used notations and descriptions in this paper are shown in **Appendix B**.

**Definition 1 (Road Network).** *The road network is a weighted graph $\mathcal{G} = (\mathcal{M}, \mathcal{E}, A)$, where $\mathcal{M} = \{m_1, m_2, \dots\}$ is the set of sensors, $\mathcal{E} \subseteq \mathcal{M} \times \mathcal{M}$ is the set of edges, and $A \in \mathbb{R}^{|\mathcal{M}| \times |\mathcal{M}|}$ is the weighted adjacency matrix of edges. Here, $m_i$ denotes the sensor with index $i$.*

**Definition 2 (Traffic Data).** *Given the available sensors $M_t = \{m_i \mid i \leq |\mathcal{M}|\}$, the traffic data is denoted as $\mathcal{X} = \{X_1, X_2, \dots\}$, where $X_t \in \mathbb{R}^{|M_t| \times F_1}$ is the traffic data of $|M_t|$ available sensors at time $t$. Here, $F_1$ denotes the number of traffic data features. For instance, $F_1 = 3$ when the traffic data includes flow, speed, and occupancy data.*

**Problem Formulation (FTT).** In federated learning, multiple clients $\mathcal{C} = \{c_1, c_2, \dots, c_n\}$ collaboratively train a global model using their local data. In the first stage, FTT trains a traffic model $\theta_{TP}$ to learn traffic knowledge from source cities $\mathcal{R} = \{R_1, R_2, \dots, R_n\}$, where each source city $R_i$ corresponds to a client $c_i$, as formally shown below:

$$\min_{\theta_{TP}} \frac{1}{n} \sum_{i=1}^{n} \mathcal{L}(\theta_{TP}, D^{R_i}), \tag{1}$$

where $\mathcal{L}(\cdot)$ is the loss function, and $D^{R_i} = \{X_1^{R_i}, X_2^{R_i}, \dots; \mathcal{G}^{R_i}\}$ is the traffic dataset of the source city $R_i$. Here, $\mathcal{G}^{R_i}$ and $X_t^{R_i}$ are the road network and the traffic data at time $t$ of the source city $R_i$. In the second stage, given target city' dataset $D^S = \{X_1^S, X_2^S, \dots; \mathcal{G}^S\}$, FTT predicts the next $T'$ traffic data based on the $T$ historical observations at time $t$ in the target city $S$, as shown below:

$$\{X_{t-T+1}^S, X_{t-T+2}^S, ..., X_t^S; \mathcal{G}^S\} \xrightarrow{\theta_{TP}} \{X_{t+1}^S, X_{t+2}^S, ..., X_{t+T'}^S\} \tag{2}$$

## 3 Our Methods

Fig. 3 illustrates the architecture of the proposed FedTT framework, which comprises three modules: Traffic View Imputation (TVI), Traffic Domain Adapter (TDA), and Traffic Secret Aggregation (TSA). As shown in Fig. 3(a), FedTT comprises $n$ clients $\mathcal{C} = \{c_1, c_2, \dots, c_n\}$ and a central server $s$. Specifically, each source city $R_i$ is treated as a client $c_i$, while the target city $S$ is treated as the server $s$. The traffic domains of the data in clients are transformed to align with the server's domain, and the server's traffic model is trained on this transformed data uploaded by clients. Consequently, the FTT problem defined in Eqs. 1 and 2 is reformulated to minimize the sum of the following losses:

$$\min_{\theta_{TP}} \frac{1}{n} \sum_{i=1}^{n} \mathcal{L}(\theta_{TP}, D^{R_i \rightarrow S}, D^S), \tag{3}$$

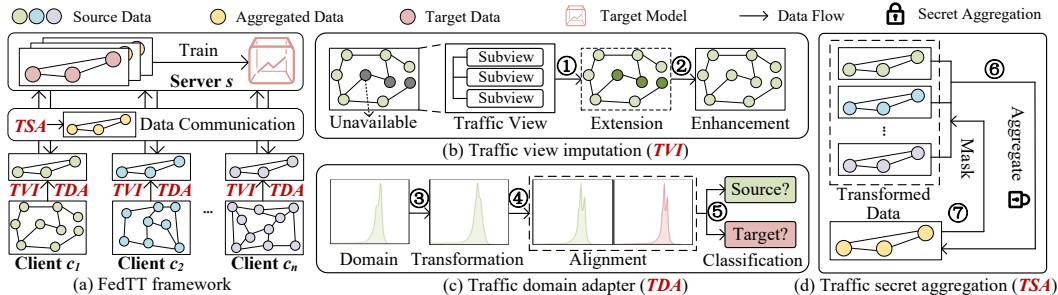

Figure 3: The architecture of the proposed FedTT framework

where $D^{R_i \to S}$ represents the traffic dataset whose domain is transformed from the source city $R_i$ to the target city $S$. The overall process of FedTDP is as following. First, the TVI module captures spatial and temporal dependencies within the traffic data to extend and enhance the traffic view (①–②), as shown in Fig. 3(b). Then, the TDA module conducts traffic domain transformation and alignment for the source cities' data (③–④). Besides, the module performs traffic domain classification to categorize the traffic data domain (⑤), as shown in Fig. 3(c). Finally, the TSA module employs the proposed traffic secret aggregation method to securely mask and aggregate the transformed data from source cities (⑥–⑦), as shown in Fig. 3(d). The target of our FedTT is to transfer traffic knowledge across cities while preserving privacy, handling data discrepancies and low data quality challenges.

## 3.1 Traffic View Imputation

**Design Motivation.** Existing federated traffic transfer methods often overlook the challenges associated with low-quality traffic data, especially when missing data is prevalent, thereby significantly undermining the performance of traffic knowledge transfer models. Although some data augmentation methods [8, 48, 73] can be leveraged for imputation, they fail to effectively capture the spatio-temporal dependencies of data, leading to suboptimal accuracy. In contrast, we propose the Traffic View Imputation (TVI) method to enhance traffic data quality by completing missing traffic data through a comprehensive exploration of the spatial and temporal dependencies inherent in traffic data:

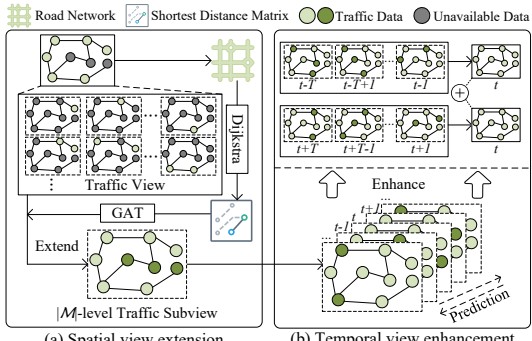

Figure 4: The process of traffic view imputation

$$\{X_1, X_2, \ldots; \mathcal{G}\} \xrightarrow{\theta_{TVI}} \{\widetilde{X}_1, \widetilde{X}_2, \ldots\}, \tag{4}$$

where $\theta_{TVI}$ is the TVI model consisting of a spatial view extension model $\theta_{SV}$ and a temporal view enhancement model $\theta_{TV}$. Besides, $\widetilde{X}_t$ is the imputed traffic data of all sensors. In addition, the traffic view represents the traffic data of all sensors at a certain time, as defined below.

**Definition 4 (Traffic View).** *A traffic view is the snapshot of traffic data of sensors $\mathcal{M}$ at time $t$, consisting of a set of multi-level traffic subviews, denoted as $V_t = \{v_t^1, v_t^2, \ldots v_t^{|M_t|}\}$, where $i$-level traffic subview $v_t^i$ is a set of traffic data of $i$ sensors at time $t$.*

**i) Spatial View Extension.** In the first stage, TVI extends the $|\mathcal{M}|$-level traffic subview at time $t$:

$$\{v_t^1, v_t^2, \ldots v_t^{|M_t|}; \mathcal{G}\} \xrightarrow{\theta_{SV}} sv_t^{|\mathcal{M}|}, \tag{5}$$

where $\theta_{SV}$ denotes the spatial view extension model and $sv_t^{|\mathcal{M}|}$ represents the extended $|\mathcal{M}|$-level traffic subview at time $t$. As shown in Fig. 4(a), it first computes the shortest distance matrix $\mathcal{A} = \{A_1, A_2, \ldots, A_{|\mathcal{M}|}\}$, where $A_i$ represents the shortest distance tensor of sensor $m_i$ to other sensors. This is computed using Dijkstra's algorithm [14] with the weighted adjacency matrix $A$. Next, the feature of each sensor is computed, i.e., $h_i = \theta_{GAT}(A_i)$, where $h_i$ represents the $K$-head feature of sensor $m_i$ with $F_2$ feature dimensions, and $\theta_{GAT}$ is the Graph Attention Network (GAT) model [61] with $K = 8$ and $F_2 = 128$. Additionally, the extension of multi-level traffic subviews is averaged to obtain the $|\mathcal{M}|$-level traffic subview with a Multi-Layer Perception (MLP [54]) $\theta_E$:

$$sv_t^{|\mathcal{M}|} = \frac{1}{|V_t|} \sum_{i=1}^{|V_t|} \frac{1}{|v_t^i|} \sum_{j=1}^{|v_t^i|} \theta_E \left( \frac{1}{i} \sum_{k=1}^{i} (H(v_t^i[j][k]) \cdot (v_t^i[j][k])^\top) \right), \tag{6}$$

where $v_t^i[j][k]$ represents the traffic data of the $k$-th sensor in the $j$-th combination within the $i$-level traffic subview at time $t$, and $H(v_t^i[j][k]) \in \mathbb{R}^{K \times F_2 \times 1}$ represents the multi-head feature of the sensor corresponding to $v_t^i[j][k]$. Finally, it computes the loss of available sensors to train the $\theta_{SV}$ model:

$$\min_{\theta_{SV}} \mathcal{L}(\theta_{SV}, \mathcal{V}_{SV}) = \min_{\theta_{SV}} \frac{1}{|\mathcal{V}_{SV}|} \sum_{t=1}^{|\mathcal{V}_{SV}|} \frac{1}{|M_t|} (sv_t^{|M_t|} - X_t), \tag{7}$$

where $\mathcal{V}_{SV} = \{sv_1^{|\mathcal{M}|}, sv_2^{|\mathcal{M}|}, \ldots\}$ is the set of extended traffic subviews at different times, and $sv_t^{|M_t|}$ is the predicted traffic data of available sensors at time $t$.

**ii) Temporal View Enhancement.** As shown in Fig. 4(b), in the second stage, TVI enhances the $|\mathcal{M}|$-level traffic subview based on the preceding/succeeding $T$ $|\mathcal{M}|$-level traffic subviews:

$$\begin{aligned} \{sv_{t-T}^{|\mathcal{M}|}, sv_{t-T+1}^{|\mathcal{M}|}, \ldots, sv_{t-1}^{|\mathcal{M}|}\} \xrightarrow{\theta_{TV}} tv_t^{|\mathcal{M}|}, \\ \{sv_{t+T}^{|\mathcal{M}|}, sv_{t+T-1}^{|\mathcal{M}|}, \ldots, sv_{t+1}^{|\mathcal{M}|}\} \xrightarrow{\theta_{TV}} tv_t^{|\mathcal{M}|}, \end{aligned} \tag{8}$$

where $tv_t^{|\mathcal{M}|}$ represents the enhanced $|\mathcal{M}|$-level traffic subview, whose final value is the average of the above two results. Besides, $\theta_{TV}$ is the temporal view enhancement model, which employs the SOTA DyHSL traffic model [80]. Then, it computes the loss of available sensors to train the $\theta_{TV}$ model:

$$\min_{\theta_{TV}} \mathcal{L}(\theta_{TV}, V^{|\mathcal{M}|}) = \min_{\theta_{TV}} \frac{1}{|V^{|\mathcal{M}|}|} \sum_{t=1}^{|V^{|\mathcal{M}|}|} \frac{1}{|M_t|} (tv_t^{|M_t|} - X_t), \tag{9}$$

where $\mathcal{V}_{TV} = \{tv_1^{|\mathcal{M}|}, tv_2^{|\mathcal{M}|}, \ldots\}$ represents the set of enhanced traffic subviews and $tv_t^{|M_t|}$ is the predicted traffic data of the available sensors at time $t$. Finally, we get the predicted traffic data of all $|\mathcal{M}|$ sensors $\widetilde{X}_t = tv_t^{|\mathcal{M}|}$. Note that the training of the TVI model is completed before the training of the FedTT framework, as it only needs to be conducted within each city.

## 3.2 Traffic Domain Adapter

**Design Motivation.** None of the existing approaches consider traffic data distribution discrepancies between the source and target cities in FTT, which decreases the effectiveness of traffic knowledge transfer. Motivated by this, to reduce the impact of traffic data distribution discrepancies on model performance, we propose the Traffic Domain Adapter (TDA) module, as shown in Fig. 5. This module reduces traffic domain discrepancies by uniformly transforming data from the traffic domain of the source city ("source domain" for short) to the traffic domain of the target city ("target domain" for short):

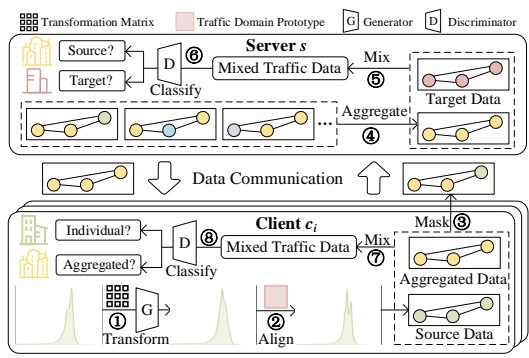

Figure 5: TDA and TSA modules

$$\{\widetilde{X}_1^R, \widetilde{X}_2^R, \ldots\} \xrightarrow{\theta_{TDA}} \{X_1^{R \to S}, X_2^{R \to S}, \ldots\}, \tag{10}$$

where $X_t^{R \to S}$ is the transformed data of $|\mathcal{M}^S|$ sensors, and $\theta_{TDA}$ is a generative adversarial network [62] consisting of a generator model $\theta_{Gen}$ and a discriminator model $\theta_{Dis}$.

**i) Traffic Domain Transformation.** In the first step, TDA uses the generator model, road network, and traffic domain prototype to transform the traffic data from the source domain to the target domain, as shown in Fig. 5 (①), where the traffic domain prototype is the representative traffic sample that can reflect the main feature of traffic data in the domain, as formally defined below.

**Definition 5 (Traffic Domain Prototype).** *Given the traffic data $\mathcal{X} = \{X_1, X_2, \ldots\}$ in a traffic domain, a traffic domain prototype $\mathcal{P}$ is the central traffic data, which is computed as the averaged value of all traffic data, i.e., $\mathcal{P} = \frac{1}{|\mathcal{X}|} \sum_{t=1}^{|\mathcal{X}|} X_t$.*

First, it computes the transformation matrix $A_{\mathcal{G}}$ of the road network through $(A_{\mathcal{G}})^\top \cdot \mathcal{G}^R \cdot A_{\mathcal{G}} = \mathcal{G}^S$, where $A_{\mathcal{G}}$ can learn the road network information of the source and target cities, which is computed by the gradient descent method [53]. Similarly, it then computes the transformation matrix $A_{\mathcal{P}}$ of the traffic domain prototype through $A_{\mathcal{P}} \cdot \mathcal{P}^R = \mathcal{P}^S$, where $\mathcal{P}^R$ and $\mathcal{P}^S$ are traffic domain prototypes of

the source and target cities, respectively. Here, $A_{\mathcal{P}}$ can learn the traffic domain prototype information of the source and target cities, which is computed by the gradient descent method. Then, the generator model leverages $A_{\mathcal{G}}$ and $A_{\mathcal{P}}$ to transform the traffic data using MLP models $\theta_{\mathcal{G}}$, $\theta_{\mathcal{P}}$, and $\theta_X$:

$$X_t^{R \to S} = \theta_{\mathcal{G}}(A_{\mathcal{G}} \cdot \widetilde{X}_t^R) + \theta_{\mathcal{P}}(A_{\mathcal{P}} \cdot \widetilde{X}_t^R) + \theta_X(\widetilde{X}_t^R), \tag{11}$$

**ii) Traffic Domain Alignment.** In the second step, TDA trains the generator model $\theta_{Gen}$, as shown in Fig. 5 (②). Specifically, it aligns the transformed data $\mathcal{X}^{R \to S} = \{\widetilde{X}_1^{R \to S}, X_2^{R \to S}, \ldots\}$ of the source city with the traffic domain prototype $\mathcal{P}^S$ of the target city $S$, as described below:

$$\min_{\theta_{Gen}} \mathcal{L}(\theta_{Gen}, \mathcal{X}^{R \to S}) = \min_{\theta_{Gen}} \frac{1}{|\mathcal{X}^{R \to S}|} \sum_{t=1}^{|\mathcal{X}^{R \to S}|} \frac{1}{|\mathcal{M}^S|} (X_t^{R \to S} - \mathcal{P}^S), \tag{12}$$

**iii) Traffic Domain Classification.** In the third step, TDA trains the discriminator model $\theta_{Dis}$ to classify the traffic data domain (⑤–⑥ shown in Fig. 5), as shown below:

$$\theta_{Dis}(X_t^{RS} \in \mathcal{X}^{RS}) = \begin{cases} P(X_t^{RS} \in \mathcal{X}^{R \to S}) \\ P(X_t^{RS} \in \mathcal{X}^S) \end{cases}, \tag{13}$$

where $\mathcal{X}^{RS} = \{X_1^{RS}, X_2^{RS}, \ldots\}$ is the traffic data mixed with the transformed data $\mathcal{X}^{R \to S}$ of the source city and the traffic data $\mathcal{X}^S$ of the target city. Besides, discriminator model $\theta_{Dis}$ is a MLP model. Then, the training process of $\theta_{Dis}$ is shown below:

$$\min_{\theta_{Dis}} \mathcal{L}(\theta_{Dis}, \mathcal{X}^{RS}) = \min_{\theta_{Dis}} \frac{1}{|\mathcal{X}^{RS}|} \sum_{t=1}^{|\mathcal{X}^{RS}|} \begin{cases} -log(P(X_t^{RS} \in \mathcal{X}^{R \to S})), & if \ X_t^{RS} \in \mathcal{X}^{R \to S} \\ -log(P(X_t^{RS} \in \mathcal{X}^S)) & , \ if \ X_t^{RS} \in \mathcal{X}^S \end{cases} \tag{14}$$

Next, we update the training process of the generator model $\theta_{Gen}$ in Eq. 12, as shown below:

$$\min_{\theta_{Gen}} \mathcal{L}(\theta_{Gen}, \theta_{Dis}, \mathcal{X}^{R \to S}, \mathcal{X}^{RS}) = \min_{\theta_{Gen}} \mathcal{L}(\theta_{Gen}, \mathcal{X}^{R \to S}) - \lambda_1 \mathcal{L}(\theta_{Dis}, \mathcal{X}^{RS}), \tag{15}$$

where $\lambda_1$ is the hyperparameter to control the trade-off between generator loss and discriminator loss.

## 3.3 Traffic Secret Aggregation

**Design Motivation.** Existing works upload gradients or models for aggregation in FTT, where attackers derive the traffic data through inference attacks [18, 67, 81]. Although techniques such as Homomorphic Encryption (HE) [52] and Differential Privacy (DP) [16] can be employed for secure aggregation, they come with notable trade-offs. Specifically, HE introduces significant computational and communication overheads, reducing training efficiency, while DP reduces the data utility, leading to lower model accuracy. In contrast, we design the Traffic Secret Aggregation (TSA) protocol that securely transmits and aggregates the transformed data from source cities to protect traffic data privacy without sacrificing the training efficiency or model accuracy, as shown in Fig. 5 (③–④).

Specifically, it first masks the $r$-th transformed data $R_i \ X_{(r)}^{R_i \to S}$ in the client $c_i$, as shown below:

$$X_{(r)}^{(\mathcal{R} \to S, \ R_i)} = \overline{X}_{(r-1)}^{\mathcal{R} \to S} + \frac{X_{(r)}^{R_i \to S} - X_{(r-1)}^{R_i \to S}}{n}, \tag{16}$$

where $\overline{X}_{(r)}^{\mathcal{R} \to S}$ is $r$-th aggregated data. Besides, $X_{(r)}^{(\mathcal{R} \to S, \ R_i)}$ is the $r$-th mask data computed in the client $c_i$ and transmitted to the server. Note that, when $r = 0$, the client uses HE to encrypt its transformed data and transmitted the encrypted data to the server for initial aggregation. Then, the server computes the sum of mask data from all source cities, as shown below:

$$\begin{aligned} \sum_{i=1}^n X_{(r)}^{(\mathcal{R} \to S, \ R_i)} &= n * \overline{X}_{(r-1)}^{\mathcal{R} \to S} + \frac{1}{n} * \sum_{i=1}^n X_{(r)}^{R_i \to S} - \frac{1}{n} * \sum_{i=1}^n X_{(r-1)}^{R_i \to S} \\ &= n * \overline{X}_{(r-1)}^{\mathcal{R} \to S} + \overline{X}_{(r)}^{\mathcal{R} \to S} - \overline{X}_{(r-1)}^{\mathcal{R} \to S} \\ &= (n-1) * \overline{X}_{(r-1)}^{\mathcal{R} \to S} + \overline{X}_{(r)}^{\mathcal{R} \to S} \end{aligned} \tag{17}$$

Finally, the server gets the $r$-th aggregated data using the previous aggregated data, as shown below:

$$\overline{\mathcal{X}}_{(r)}^{\mathcal{R} \to S} = \sum_{i=1}^n \mathcal{X}_{(r)}^{(\mathcal{R} \to S, \ R_i)} - (n-1) * \overline{\mathcal{X}}_{(r-1)}^{\mathcal{R} \to S} \tag{18}$$

In this way, it ensures that only the aggregated data can be accessed without revealing the individual transformed data. Besides, the client $c_i$ can train a local discriminator model $\theta_{Dis}^{R_i}$ to classify the aggregated data and individual transformed data (⑦–⑧ shown in Fig. 5), as shown below:

$$\theta_{Dis}^{R_i}(X_t^{R_iS} \in \mathcal{X}^{R_iS}) = \begin{cases} P(X_t^{R_iS} \in \mathcal{X}^{R_i \to S}) \\ P(X_t^{R_iS} \in \overline{\mathcal{X}}^{\mathcal{R} \to S}) \end{cases}, \tag{19}$$

where $\mathcal{X}^{R_iS} = \{X_1^{R_iS}, X_2^{R_iS}, \ldots\}$ is the traffic data mixed with the aggregated data $\overline{\mathcal{X}}^{\mathcal{R} \to S}$ and transformed data $\mathcal{X}^{R_i \to S}$. Besides, $\theta_{Dis}^{R_i}$ is a MLP model and its training process is shown below:

$$\min_{\theta_{Dis}^{R_i}} \mathcal{L}(\theta_{Dis}^{R_i}, \mathcal{X}^{R_iS}) = \min_{\theta_{Dis}^{R_i}} \frac{1}{|\mathcal{X}^{R_iS}|} \sum_{t=1}^{|\mathcal{X}^{R_iS}|} \begin{cases} -log(P(X_t^{R_iS} \in \mathcal{X}^{R_i \to S})), \ if \ X_t^{R_iS} \in \mathcal{X}^{R_i \to S} \\ -log(P(X_t^{R_iS} \in \mathcal{X}^{\mathcal{R} \to S})), \ if \ X_t^{R_iS} \in \mathcal{X}^{\mathcal{R} \to S} \end{cases} \tag{20}$$

Therefore, given the traffic data $\mathcal{X}^{\mathcal{R}S} = \{X_1^{\mathcal{R}S}, X_2^{\mathcal{R}S}, \ldots\}$ consisting of aggregated data $\overline{\mathcal{X}}^{\mathcal{R} \to S}$ and traffic data $\mathcal{X}^S$, the updated training process of the generator model $\theta_{Gen}$ in Eq. 15 is shown below:

$$\min_{\theta_{Gen}^{R_i}} \mathcal{L}(\theta_{Gen}^{R_i}, \mathcal{X}^{R_i \to S}) - \lambda_1 \mathcal{L}(\theta_{Dis}, \mathcal{X}^{\mathcal{R}S}) - \lambda_2 \mathcal{L}(\theta_{Dis}^{R_i}, \mathcal{X}^{R_iS}), \tag{21}$$

where $\theta_{Gen}^{R_i}$ and $\theta_{Dis}$ are the local generator model and global discriminator model in the client $c_i$ and server $s$, respectively. Here, $\lambda_1$ and $\lambda_2$ are the hyperparameter to control the trade-off between generator loss and discriminator loss.

The overall training process and theoretical privacy analysis of FedTT are shown in **Appendix C**.

## 4   Experiment

Table 1: Statistics of evaluated datasets

| Dataset | # instances | # sensors | Interval | City | Missing Rate |
|---------|-------------|-----------|----------|------|--------------|
| PeMSD4 | 16992 | 307 | 5 min | San Francisco | 16.35% |
| PeMSD8 | 17856 | 170 | 5 min | San Bernardino | 20.09% |
| FT-AED | 1920 | 196 | 5 min | Nashville | 4.59% |
| HK-Traffic | 17856 | 411 | 5 min | Hong Kong | 13.01% |

**Datasets.** We use four traffic datasets to evaluate the proposed FedTT framework in experiments, which are widely used in traffic prediction tasks [80, 23, 24], as shown in Table 1. Specifically, PeMSD4 (**P4**) [3], PeMSD8 (**P8**) [3], FT-AED (**FT**) [12], and HK-Traffic (**HK**) [2] were collected in the San Francisco, San Bernardino, Nashville, and Hong Kong, respectively. Among them, three datasets are considered as three source cities, and one dataset serves as the target city, leading to four scenarios: (P8, FT, HK) → P4, (P4, FT, HK) → P8, (P4, P8, HK) → FT, and (P4, P8, FT) → HK. Besides, we select traffic flow, speed, and occupancy prediction tasks for experiments, which are also widely studied in the community [80, 23, 24]. In addition, we report the rate of missing traffic data in these datasets, which reveals varying levels of traffic data quality issues.

**Baselines.** We compare FedTT with (i) three SOTA **methods in FTT** including T-ISTGNN [49], pFedCTP [78], and 2MGTCN [75], (ii) three SOTA **Multi-Source Traffic Knowledge Transfer methods (MTT)** extended for the FTT problem including TPB [41], ST-GFSL [43], and DastNet [57], and (iii) three SOTA **Single-Source Traffic Knowledge Transfer methods (STT)** for the FTT problem including CityTrans [47], TransGTR [27], and MGAT [46]. In addition, we replace the TVI module of FedTT with three SOTA data imputation methods (LATC [8], GCASTN [48], and Nuhuo [73]) to evaluate its effects. More details about these baselines are provided in **Appendix D.1**.

**Evaluation Metrics.** We use Mean Absolute Error (MAE), Root Mean Square Error (RMSE), communication size (GB), and running time (minutes) to evaluate the utility in experiments. Besides, Mean Square Error (MSE) and Pearson Correlation Coefficient (PCC) between the reconstructed data and the ground truth data to measure the privacy-preserving ability of different methods.

**Implementation.** All baselines run under their optimal settings. Besides, we use 5% train data, 10% validation data, and 10% test data in the target city. In addition, the MLP model used in FedTT is three-layer with the GELU [21] activation and 1024 hidden dimensions. Moreover, all experiments are conducted with four nodes, one as a server and the other three nodes as clients, each equipped with two Intel Xeon CPU E5-2650 12-core processors and two NVIDIA GeForce RTX 3090.

Table 2: The overall performance comparison between different methods

| Metric | Method | (P8, FT, HK) → P4[1] | | | (P4, FT, HK) → P8 | | | (P4, P8, HK) → FT | | | (P4, P8, FT) → HK | | |
|---|---|---|---|---|---|---|---|---|---|---|---|---|---|
| | | flow | speed | occ | flow | speed | occ | flow | speed | occ | flow | speed | occ |
| MAE | 2MGTCN | 20.34 | 1.27 | 0.0077 | 16.39 | 1.09 | 0.0069 | 13.86 | 4.77 | 0.0355 | 8.49 | 1.38 | 0.0094 |
| | pFedCTP | 21.24 | 1.52 | 0.0079 | 17.06 | 1.22 | 0.0072 | 13.92 | 5.78 | 0.0415 | 9.22 | 1.22 | 0.0102 |
| | T-ISTGNN | 27.24 | 2.03 | 0.0219 | 22.75 | 1.84 | 0.0235 | 20.83 | 9.69 | 0.0571 | 9.98 | 4.24 | 0.0121 |
| | TPB | 21.06 | 1.28 | 0.0134 | 17.11 | 1.12 | 0.0081 | 13.03 | 3.59 | 0.0276 | 8.36 | 1.52 | 0.0092 |
| | ST-GFSL | 23.05 | 1.47 | 0.0161 | 19.86 | 1.47 | 0.0159 | 18.00 | 5.25 | 0.0385 | 8.42 | 2.03 | 0.0101 |
| | DastNet | 26.89 | 1.54 | 0.0165 | 19.58 | 1.41 | 0.0134 | 15.44 | 4.62 | 0.0421 | 9.09 | 3.85 | 0.0135 |
| | CityTrans | 23.94 | 1.38 | 0.0119 | 18.51 | 1.18 | 0.0108 | 13.06 | 3.60 | 0.0359 | 8.78 | 1.84 | 0.0116 |
| | TransGTR | 24.32 | 1.39 | 0.0135 | 19.53 | 1.18 | 0.0089 | 13.27 | 4.80 | 0.0337 | 9.09 | 3.92 | 0.0102 |
| | MGAT | 24.78 | 1.58 | 0.0195 | 20.16 | 1.67 | 0.0160 | 20.08 | 8.00 | 0.0469 | 9.14 | 2.88 | 0.0101 |
| | **FedTT** | **16.69** | **1.03** | **0.0061** | **14.11** | **0.94** | **0.0059** | **12.10** | **3.24** | **0.0249** | **7.42** | **1.05** | **0.0087** |
| RMSE | 2MGTCN | 31.61 | 2.27 | 0.0179 | 25.95 | 2.18 | 0.0131 | 17.03 | 7.49 | 0.0644 | 12.11 | 3.25 | 0.00167 |
| | pFedCTP | 33.03 | 3.12 | 0.0188 | 26.19 | 2.62 | 0.0164 | 19.94 | 9.84 | 0.0756 | 13.31 | 2.62 | 0.0212 |
| | T-ISTGNN | 35.95 | 4.14 | 0.0281 | 31.10 | 3.37 | 0.0305 | 29.42 | 13.17 | 0.1127 | 15.68 | 6.31 | 0.0230 |
| | TPB | 31.75 | 2.31 | 0.0201 | 26.35 | 2.19 | 0.0126 | 16.34 | 6.07 | 0.0493 | 11.89 | 2.98 | 0.0152 |
| | ST-GFSL | 33.65 | 3.29 | 0.0237 | 30.66 | 3.12 | 0.0260 | 22.10 | 9.69 | 0.0652 | 12.89 | 4.73 | 0.0156 |
| | DastNet | 34.96 | 3.41 | 0.0274 | 27.45 | 3.10 | 0.0299 | 22.64 | 9.72 | 0.0691 | 13.63 | 5.82 | 0.0236 |
| | CityTrans | 32.04 | 2.46 | 0.0237 | 27.91 | 2.20 | 0.0226 | 18.86 | 9.82 | 0.0514 | 13.45 | 4.72 | 0.0212 |
| | TransGTR | 33.66 | 2.43 | 0.0198 | 26.41 | 2.27 | 0.0147 | 17.11 | 7.96 | 0.0579 | 12.23 | 6.77 | 0.0180 |
| | MGAT | 32.85 | 3.43 | 0.0283 | 30.77 | 3.20 | 0.0262 | 24.62 | 11.05 | 0.1028 | 12.03 | 5.11 | 0.0162 |
| | **FedTT** | **27.48** | **1.93** | **0.0166** | **24.29** | **1.94** | **0.0099** | **15.91** | **5.50** | **0.0372** | **8.57** | **2.40** | **0.0145** |

[1] P4, P8, FT, and HK denote PeMSD4, PeMSD8, FT-AED, and HK-Traffic datasets, respectively.

## 4.1 Overall Performance

To show the overall performance of different methods on traffic flow, speed, and occupancy ("occ" for short) predictions tasks, we take 60 minutes (12-time steps) of historical data as input and output the traffic prediction in the next 15 minutes (3-time steps), as shown in Table 2, where the best results are shown in blue. Here, the DyHSL [80] model is implemented in FedTT as it achieves the state-of-the-art performance in the centralized traffic model. As observed, the proposed FedTT framework achieves the best performance on different traffic datasets and traffic prediction tasks compared to other methods, showing its effectiveness of traffic knowledge transfer in the FTT problem, i.e., the gains range from **5.43% to 75.24%** in MAE and **2.63% to 67.54%** in RMSE.

## 4.2 Privacy Protection Study

To evaluate the privacy-preserving capabilities, we conduct the data reconstruction attack to different methods across datasets on traffic flow prediction using MSE and PCC, as illustrated in Fig. 6. As observed, FedTT demonstrates robust resistance to the data reconstruction attack, achieving a high MSE and maintaining a PCC within **2.17% to 8.81%**, not exceeding 10%, while other methods exhibit weaker defenses, with a lower MSE and PCC larger than 40%.

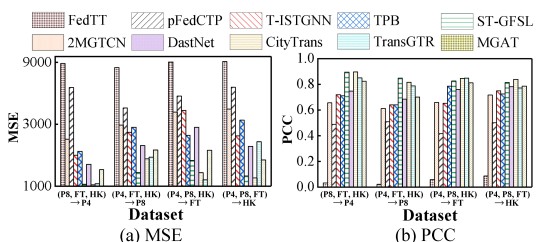

Figure 6: Privacy protection study

These findings underscore the superiority and effectiveness of privacy protection provided by the proposed FedTT framework in FTT and highlight the limitations of privacy preservation mechanisms based solely on traditional federated learning frameworks.

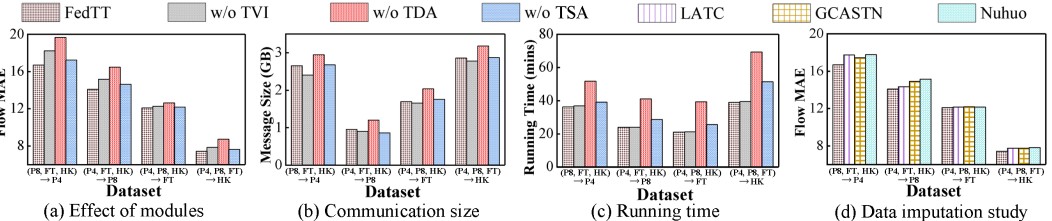

Figure 7: Ablation study of FedTT

## 4.3 Ablation Study

Fig. 7 shows the ablation study, where we removed the module of FedTT one at a time, namely FedTT without TVI (w/o TVI), FedTT without TDA (w/o TDA), and FedTT without TSA (w/o TSA). First, when TVI is absent, MAE increases by **1.49% to 9.23%**, underscoring its pivotal role as an effective way to complete the missing data. Besides, the training of TVI is completed before the FedTT's training as it only needs to be conducted within each source city, thus not increasing communication

overhead or running time during FedTT's training. Additionally, compared to other data imputation methods (i.e., LATC, GCASTN, and Nuhuo), FedTT with TVI achieves better performance, showing its effectiveness in the traffic data completion. Second, when TDA is removed, MAE increases by **4.46% to 17.86%**, which demonstrates its effectiveness in addressing traffic data distribution differences. Besides, communication overhead and running time of FedTT slightly increase compared to w/o TDA. Third, MAE of FedTT decreases **0.66% to 3.76%** compared to w/o TSA as TSA uses the averaged source data, which reduces the influence of source city's traffic patterns on the target city's model training. Besides, the communication overhead and running time of FedTT compared to w/o TSA do not change as TSA is a lightweight module for federated secure aggregation.

### 4.4 Long-Term Traffic Prediction

To evaluate long-term traffic prediction capabilities, we illustrate the performance of different methods over the next 60 minutes (12 time steps) for traffic flow and speed prediction using MAE, as shown in Fig. 8. As observed, FedTT outperforms all other methods, i.e., the gains range from **5.03% to 64.41%**, showing its effectiveness of long-term traffic prediction in FTT. Therefore, the proposed FedTT framework demonstrates strong performance in both

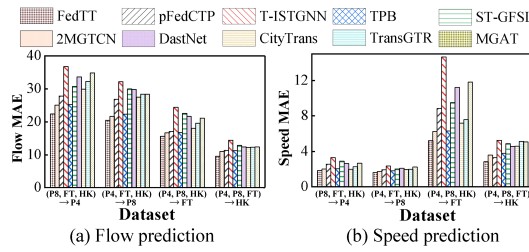

(a) Flow prediction    (b) Speed prediction

Figure 8: Long-term traffic prediction

long-term and short-term traffic prediction (i.e., Table 2), underscoring its general advantages in FTT.

### 4.5 Model Scalability

To validate the model scalability, we show the traffic flow and speed prediction performance of different methods across different sizes of training data in the target city, ranging from 5% to 40% in the (P8, FT, HK) → P4 scenario using MAE, as shown in Fig. 9. As observed, the FedTT framework consistently achieves the best performance in different-scale datasets with **7.22% to 49.26%** MAE less than other methods, indicating its superior scalability in FTT.

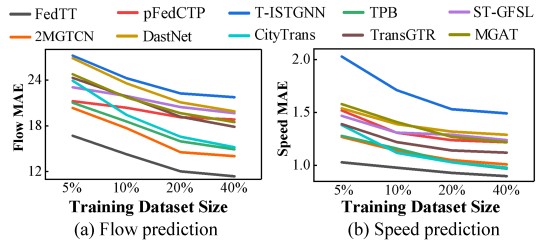

(a) Flow prediction    (b) Speed prediction

Figure 9: Model scalability study

Besides, as the size of the training data increases, all methods exhibit improved performance. This is because more training data enhances the model learning capability on the target city's traffic pattern.

### 4.6 More Experiments

We conduct more experiments to comprehensive evaluate FedTT, in terms of model adaptability, efficiency, hyperparameter sensitivity, and case study: i) **Appendix D.2** demonstrates the performance when extending different centralized traffic models to FedTT and the two-stage transfer of existing methods in FTT, where FedTT achieves 5.13% to 64.65% lower MAE in all models. ii) **Appendix D.3** shows the efficiency of different methods, where FedTT reduces communication overhead by 90% and running time by 1 to 2 orders of magnitude compared to all baselines. iii) **Appendix D.4** shows the FedTT's performance with different hyperparameter settings, where $\lambda_1 = 0.7$ and $\lambda_2 = 0.4$ are optimum values. iv) **Appendix D.5** showcases FedTT's practical efficacy in a real-world scenario.

## 5 Conclusion and Limitations

In this paper, we propose FedTT, a privacy-aware and efficient federated learning framework for cross-city traffic knowledge transfer. It includes a traffic view imputation method to enhance data quality, a traffic domain adapter to address data distribution discrepancies, and a traffic secret aggregation protocol to safeguard data privacy. Experiments using 4 datasets demonstrate its superiority. Our work has several limitations that warrant further exploration. First, we have not addressed grid-based scenarios, which could be an important direction for future research. Besides, while our study primarily focuses on traffic prediction tasks, extending the framework to support more spatio-temporal prediction tasks remains an open opportunity. In addition, we have not systematically evaluated the impact of varying the number of source cities on the performance of traffic knowledge transfer, which could provide additional insights into the scalability of the proposed framework.

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

543 # Appendix

544

# Appendix

In the subsequent sections, we present supplementary materials to provide more details of this paper, offering deeper insights and additional technical details for readers seeking further clarification. The appendix is organized as follows.

In **Section A**, we present a systematic review of related work to help readers understand the key development in areas relevant to this paper, including traffic prediction and traffic knowledge transfer.

In **Section B**, we summary the frequently used notations and descriptions for better understanding our work.

In **Section C**, we provide the additional methodology details of our proposed FedTT framework, including (i) the federated parallel training strategy, (ii) the training process with training algorithm and complexity analysis, and (iii) the theoretical privacy analysis.

In **Section D**, we describe the extensive experimental details to provide more information about experimental settings and further demonstrate the superiority performance of the proposed FedTT framework, including (i) compared baselines introduction, (ii) the details experimental results of model adaptability, efficiency, hyperparameter sensitivity, and case studies.

## A  Related Work

### A.1  Traffic Prediction

Traffic prediction plays a critical role in the development of smart cities and has garnered significant attention in the spatio-temporal data mining community. Currently, deep learning techniques [54] are widely employed in traffic prediction tasks. Convolutional models, such as Convolutional Neural Networks (CNN) [33] and Graph Convolutional Networks (GCN) [29], are used to capture spatial correlations in traffic time-series data. Meanwhile, sequential models including Gated Recurrent Units (GRU) [11] and Long Short-Term Memory (LSTM) [19], are employed to extract temporal dependencies from the data. Several advanced models have achieved state-of-the-art performance. For instance, ST-SSL [23] improves traffic pattern representation to account for spatial and temporal heterogeneity through a self-supervised learning framework. DyHSL [80] leverages hypergraph structure information to model the dynamics of a traffic network, updating the representation of each node by aggregating messages from associated hyperedges. Additionally, PDFormer [24] introduces a spatial self-attention module to capture dynamic spatial dependencies and a flow-delay-aware feature transformation module to model the time delays in spatial information propagation. Since this paper is not intended to propose another more complex prediction model, a detailed analysis of existing traffic prediction models can be found in surveys [25, 28]. However, these models are centralized and rely on traffic data uploads from sensors to a central server, which poses a risk of data leakage.

To address data privacy concerns, several traffic prediction studies [74, 31, 69, 34, 70, 38] in federated environments have been proposed. Specifically, FedGRU [39] pioneers the integration of GRU into FL for TP tasks, employing federated averaging to aggregate models and a joint announcement protocol to enhance model scalability. Subsequently, CNFGNN [45] separates the modeling of temporal dynamics on the device from spatial dynamics on the server, using alternating optimizations to reduce communication costs and facilitate computation on edge devices. Moreover, FedGTP [70] promotes the adaptive exploitation of inter-client spatial dependencies to enhance prediction performance while ensuring data privacy. However, urban traffic data is often insufficient or unavailable, particularly in emerging cities. Training traffic models in these data-scarce cities is prone to overfitting, which undermines model performance and affects the accuracy of TP tasks. ***In contrast, we aim to propose a federated traffic prediction framework that efficiently transfers traffic knowledge from data-rich cities to data-scarce cities, enhancing TP capabilities for the latter.***

### A.2  Traffic Knowledge Transfer

Transfer learning can enhance the traffic model capabilities of data-scarce target cities by transferring traffic knowledge from data-rich source cities in traffic prediction tasks. Existing studies can be broadly categorized into three types: Single-Source Traffic Knowledge Transfer (STT), Multi-

Source Traffic Knowledge Transfer (MTT), and Federated Traffic Knowledge Transfer (FTT), in chronological order from earliest to most recent.

First, STT [26, 17, 63, 36, 72, 22, 35, 9, 65] studies focus on transferring traffic knowledge from a single source city to a target city. Specifically, TransGTR [27] jointly learns transferable structure generators and forecasting models across cities to enhance prediction performance in data-scarce target cities. Next, CityTrans [47] leverages adaptive spatio-temporal knowledge and domain-invariant features for accurate traffic prediction in data-scarce cities. Additionally, MGAT [46] uses a meta-learning algorithm to extract multi-granular regional features from each source city to improve the effectiveness of traffic knowledge transfer. However, the performance of these STT methods can be significantly compromised when there are substantial differences in traffic data distribution between the source and target cities.

Second, MTT [71, 40, 78, 77] studies the joint transfer of traffic knowledge from multiple source cities to a target city, enabling the target city to acquire diverse traffic knowledge and enhancing the robustness of the trained traffic models. Specifically, TPB [41] uses a traffic patch encoder to create a traffic pattern bank, which data-scarce cities query to establish relationships, aggregate meta-knowledge, and construct adjacency matrices for future traffic prediction. Next, ST-GFSL [43] transfers knowledge through parameter matching to retrieve similar spatio-temporal features and defines graph reconstruction loss to guide structure-aware learning. Additionally, DastNet [57] employs graph representation learning and domain adaptation techniques to create domain-invariant embeddings for traffic data. However, these methods rely on centralized frameworks, which involves sharing and exchanging traffic data across cities without considering traffic data privacy.

Third, the latest FTT studies, including T-ISTGNN [49], pFedCTP [78], and 2MGTCN [75], intend to protect data privacy in cross-city traffic knowledge transfer. Specifically, T-ISTGNN [49] combines privacy-preserving traffic knowledge transfer with inductive spatio-temporal GNNs for cross-region traffic prediction. Besides, pFedCTP [78] employs personalized FL to decouple the ST-Net into shared and private components, addressing the spatial and temporal heterogeneity. In addition, 2MGTCN [75] combines multi-modal GCNs and TCNs to capture spatial and temporal information and enhance adaptability across cities. However, they face challenges such as privacy leakage, data distribution discrepancies, low data quality, and high knowledge transfer overhead, making them unsuitable for real-world applications, as shown in Fig 2. ***In contrast, we aim to propose a privacy-preserving and efficient federated learning framework for cross-city traffic knowledge transfer to address the challenges of privacy, effectiveness, robustness, and efficiency in FTT.***

## B  Notations and Descriptions

We present the frequently used notations and descriptions in this paper, as listed in Table 3.

Table 3: Notations and descriptions

| Notation | Description |
|---|---|
| $m, \mathcal{M}$ | A sensor and a set of sensors $\{m_1, m_2, \ldots\}$ |
| $\mathcal{E}, A$ | A set of edges and the weighted adjacent matrix of edges |
| $\mathcal{G}$ | A road network $(\mathcal{M}, \mathcal{E}, A)$ |
| $t, r, tr$ | The time, $r$-th, and training round |
| $M_t$ | A set of available sensors $\{m_i \mid i \leq |\mathcal{M}|\}$ at time $t$ |
| $X_t, X_{(r)}$ | The traffic data at time $t$ and the $r$-th traffic data |
| $F_1$ | The dimension of the traffic data features |
| $\mathcal{X}, D$ | A set of traffic data $\{X_1, X_2, \ldots\}$ and a traffic dataset $\{X_1, X_2, \ldots; \mathcal{G}\}$ |
| $c, s$ | A client and the server |
| $R, S$ | A source city and the target city |
| $n$ | The number of clients and source cities |
| $\mathcal{C}, \mathcal{R}$ | A set of clients $\{c_1, c_2, \ldots, c_n\}$ and source cities $\{R_1, R_2, \ldots, R_n\}$ |
| $\theta, \mathcal{L}(\cdot)$ | A model and a loss function |
| $v_t^i, V_t$ | The $i$-level traffic subview and a traffic view $\{v_t^1, v_t^2, \ldots\}$ at time $t$ |
| $\mathcal{P}$ | A traffic domain prototype |

## C Methodology Details

### C.1 Federated Parallel Training

To improve the training efficiency, FedTT introduces the federated parallel training strategy to reduce the data transmission and train the models in parallel.

**i) Split Learning.** To reduce the communication overhead and improve the training efficiency, it employs split learning [45] to decompose the sequential training process into the client and server training, and freeze the data required by the client and server. Specifically, the client $c_i$ stores and freezes the data sent by the server for $\theta_{Gen}^{R_i}$ and $\theta_{Dis}^{R_i}$ training in Eqs. 21 and Eqs. 20, respectively:

$$\min_{\theta_{Gen}^{R_i}} \mathcal{L}(\theta_{Gen}^{R_i}, \mathcal{X}^{R_i \to S}) - \lambda_1 * Fr(\mathcal{L}(\theta_{Dis}, \mathcal{X}^{\mathcal{R}S})) - \lambda_2 \mathcal{L}(\theta_{Dis}^{R_i}, \mathcal{X}^{R_i S}), \tag{22}$$

$$\min_{\theta_{Dis}^{R_i}} \frac{1}{|\mathcal{X}^{R_i S}|} \sum_{t=1}^{|\mathcal{X}^{R_i S}|} \begin{cases} -log(P(X_t^{R_i S} \in \mathcal{X}^{R_i \to S})), if \ X_t^{R_i S} \in \mathcal{X}^{R_i \to S} \\ -log(P(X_t^{R_i S} \in \mathcal{X}^{\mathcal{R} \to S})), if \ X_t^{R_i S} \in Fr(\mathcal{X}^{\mathcal{R} \to S}) \end{cases}, \tag{23}$$

where $Fr(\cdot)$ is the frozen function and uses the historical cached data, which updates every 5 rounds. Besides, the server $s$ stores and freezes the data uploaded by the client to compute the aggregated data for $\theta_{Dis}$ and traffic model $\theta_{TP}$ training in Eqs. 14 and 3, respectively:

$$\min_{\theta_{Dis}} \frac{1}{|\mathcal{X}^{\mathcal{R}S}|} \sum_{t=1}^{|\mathcal{X}^{\mathcal{R}S}|} \begin{cases} -log(P(X_t^{\mathcal{R}S} \in \mathcal{X}^{\mathcal{R} \to S})), \ if \ X_t^{\mathcal{R}S} \in Fr(\mathcal{X}^{\mathcal{R} \to S}) \\ -log(P(X_t^{\mathcal{R}S} \in \mathcal{X}^{S})) \quad , \ if \ X_t^{\mathcal{R}S} \in \mathcal{X}^{S} \end{cases}, \tag{24}$$

$$\min_{\theta_{TP}} \mathcal{L}(\theta_{TP}, Fr(D^{\mathcal{R} \to S}), D^{S}) \tag{25}$$

**ii) Parallel Optimization.** To further improve the training parallelism, it proposes parallel optimization to reduce data dependencies on the client and server. Specifically, the client $c_i$ caches and freezes the local data for $\theta_{Gen}^{R_i}$ and $\theta_{Dis}^{R_i}$ parallel training in Eqs 22 and 23, as shown below:

$$\min_{\theta_{Gen}^{R_i}} \mathcal{L}(\theta_{Gen}^{R_i}, \mathcal{X}^{R_i \to S}) - \lambda_1 * Fr(\mathcal{L}(\theta_{Dis}, \mathcal{X}^{\mathcal{R}S})) - \lambda_2 * Fr^{'}(\mathcal{L}(\theta_{Dis}^{R_i}, \mathcal{X}^{R_i S})), \tag{26}$$

$$\min_{\theta_{Dis}^{R_i}} \frac{1}{|\mathcal{X}^{R_i S}|} \sum_{t=1}^{|\mathcal{X}^{R_i S}|} \begin{cases} -log(P(X_t^{R_i S} \in \mathcal{X}^{R_i \to S})), if \ X_t^{R_i S} \in Fr^{'}(\mathcal{X}^{R_i \to S}) \\ -log(P(X_t^{R_i S} \in \mathcal{X}^{\mathcal{R} \to S})), if \ X_t^{R_i S} \in Fr(\mathcal{X}^{\mathcal{R} \to S}) \end{cases}, \tag{27}$$

where $Fr^{'}(\cdot)$ is the frozen function and uses the historical cached data, which updates each round.

### C.2 Training Process

Before the training of the FedTT framework, clients (i.e., source cities) train the spatial view expansion model $\theta_{SV}$ and the temporal view expansion model $\theta_{TV}$ in the TVI module $\theta_{TVI}$ by minimizing the loss in Eqs. 7 and 9, as shown below:

$$\min_{\theta_{TVI}} \mathcal{L}(\theta_{TVI}, \mathcal{V}_{SV}, \mathcal{V}_{TV}) = \min_{\theta_{SV}} \mathcal{L}(\theta_{SV}, \mathcal{V}_{SV}) + \min_{\theta_{TV}} \mathcal{L}(\theta_{TV}, \mathcal{V}_{TV}), \tag{28}$$

where $\mathcal{V}_{SV}$ and $\mathcal{V}_{TV}$ are the set of traffic subviews at different times obtained by spatial view extension and temporal view enhancement, respectively. During the training of the FedTT framework, the client $c_i$ trains the local generator model $\theta_{Gen}^{R_i}$ and the local discriminator model $\theta_{Dis}^{R_i}$ by minimizing the loss in Eqs. 20 and 21, as shown below:

$$\min_{\theta_{Gen}^{R_i}} \mathcal{L}(\theta_{Gen}^{R_i}, \theta_{Dis}, \theta_{Dis}^{R_i}, \mathcal{X}^{R_i \to S}, \mathcal{X}^{\mathcal{R}S}, \mathcal{X}^{R_i S}) + \min_{\theta_{Dis}^{R_i}} \mathcal{L}(\theta_{Dis}^{R_i}, \mathcal{X}^{R_i S}), \tag{29}$$

where $\mathcal{X}^{\mathcal{R}S}$ is the traffic data consisting of the aggregated data $\overline{\mathcal{X}}^{\mathcal{R} \to S}$ and traffic data $\mathcal{X}^{S}$ of the target city $S$, and $\mathcal{X}^{R_i S}$ is the traffic data consisting of the aggregated data $\overline{\mathcal{X}}^{\mathcal{R} \to S}$ and transformed data $\mathcal{X}^{R_i \to S}$ of the source city $R_i$. Besides, the server $s$ trains the global discriminator model $\theta_{Dis}$ and traffic model $\theta_{TP}$ by minimizing the loss in Eqs. 14 and 3, as shown below:

$$\min_{\theta_{Dis}} \mathcal{L}(\theta_{Dis}, \mathcal{X}^{\mathcal{R}S}) + \min_{\theta_{TP}} \mathcal{L}(\theta_{TP}, \overline{D}^{\mathcal{R} \to S}, D^{S}), \tag{30}$$

where $\overline{D}^{\mathcal{R}\to S}$ is the aggregated traffic dataset whose traffic domain is transformed from source cities to the target city $S$, and $D^S$ is the traffic dataset of the target city $S$.

---

**Algorithm 1** The training of the FedTT framework in the client $c_i$

---

1: **Input:** the server $s$ (i.e., the target city $S$).
2: $\widetilde{\mathcal{X}}^{R_i} \leftarrow Complete(\theta_{TVI}, \mathcal{X}^{R_i})$ // Complete the missing data.
3: **for** each training round $tr = 1, 2, \ldots$ **do**
4:     **for** each data $X_{(r)}^{R_i} \in \widetilde{\mathcal{X}}^{R_i}, r = 1, 2, \ldots$ **do**
5:         $X_{(r)}^{R_i\to S} \leftarrow Transform(\theta_{Gen}^{R_i}, X_{(r)}^{R_i})$ // Transform the traffic data.
6:         $Classify(\theta_{Dis}^{R_i}, X_{(r)}^{R_i\to S})$ // Classify the transformed data.
7:         **if** $tr == 1$ and $r == 1$ **then**
8:           $E_{(r)}^{R_i\to S} \leftarrow Encrypt(X_{(r)}^{R_i\to S})$ // Encrypt the transformed data.
9:           $Send(s, E_{(r)}^{R_i\to S})$ // Send the transformed data.
10:         **else**
11:           **if** $tr == 1$ and $r == 2$ **then**
12:             $\overline{E}_{(r-1)}^{\mathcal{R}\to S} \leftarrow Get(s, r)$ // Get the encrypted aggregated data.
13:             $\overline{X}_{(r-1)}^{\mathcal{R}\to S} \leftarrow Decrypt(\overline{E}_{(r-1)}^{\mathcal{R}\to S})$ // Decrypt the aggregated data.
14:           **else**
15:             $\overline{X}_{(r-1)}^{\mathcal{R}\to S} \leftarrow Get(s, r)$ // Get the aggregated data.
16:           **end if**
17:           $Classify(\theta_{Dis}^{R_i}, \overline{X}_{(r-1)}^{\mathcal{R}\to S})$ // Classify the aggregated data.
18:           $X_{(r)}^{(\mathcal{R}\to S, R_i)} \leftarrow \overline{X}_{(r-1)}^{\mathcal{R}\to S} + X_{(r)}^{R_i\to S} - X_{(r-1)}^{R_i\to S}$ // Mask the transformed data.
19:           $Send(s, X_{(r)}^{(\mathcal{R}\to S, R_i)})$ // Send the mask data.
20:         **end if**
21:     **end for**
22: **end for**

---

**Algorithm 2** The training of the FedTT framework in the server $s$

---

1: **Input:** clients $\mathcal{C} = \{c_1, c_2, \ldots, c_n\}$ (i.e., source cities $\mathcal{R} = \{R_1, R_2, \ldots, R_n\}$).
2: **for** each training round $tr = 1, 2, \ldots$ **do**
3:     **for** $r = 1, 2, \ldots$ **do**
4:         **if** $tr == 1$ and $r == 1$ **then**
5:           $\{E_{(r)}^{R_1\to S}, E_{(r)}^{R_2\to S}, \ldots\} \leftarrow Get(\mathcal{C}, r)$ // Get the encrypted data.
6:           $\overline{E}_{(r)}^{\mathcal{R}\to S} \leftarrow \sum_{i=1}^{n} E_{(r)}^{R_i\to S}$ // Aggregate the encrypted data.
7:           $Send(\mathcal{C}, \overline{E}_{(r)}^{\mathcal{R}\to S})$ // Send the aggregated data.
8:         **else**
9:           $\{X_{(r)}^{(\mathcal{R}\to S, R_1)}, X_{(r)}^{(\mathcal{R}\to S, R_2)}, \ldots\} \leftarrow Get(\mathcal{C}, r)$ // Get the mask data.
10:           $\overline{X}_{(r)}^{\mathcal{R}\to S} \leftarrow \sum_{i=1}^{n} X_{(r)}^{(\mathcal{R}\to S, R_i)} - (n-1) * \overline{X}_{(r-1)}^{\mathcal{R}\to S}$ // Aggregate the mask data.
11:           $Classify(\theta_{Dis}, \overline{X}_{(r)}^{\mathcal{R}\to S})$ // Classify the aggregated data.
12:           $Send(\mathcal{C}, \overline{X}_{(r)}^{\mathcal{R}\to S})$ // Send the aggregated data.
13:         **end if**
14:     **end for**
15:     $Classify(\theta_{Dis}, \mathcal{X}^S)$ // Classify the local data.
16:     $Prediction(\theta_{TP}, \overline{\mathcal{X}}^{\mathcal{R}\to S}, \mathcal{X}^S)$ // Perform traffic prediction.
17: **end for**

---

**Training Algorithm**. For convenient method reproduction, we provide detailed training Algorithms 1 and 2 of the FedTT framework, including the client and server.

In the client (i.e., Algorithm 1), the target city acts as the server (line 1). Before the training process, the client completes the missing traffic data through the traffic view imputation method (line 2). During each training round and each traffic data (lines 3–4), it first transforms the data from the traffic domain of the source city to that of the target city (line 5) and classifies the transformed data using the local discriminator model(line 6). If the training process is in the first round using the first data instance (line 7), the client encrypts the transformed data using homomorphic encryption and sends it to the server (lines 8-9). Otherwise, if the training process is in the first round using the second data instance (lines 10-11), the client gets the encrypted data and decrypts it to get the previous aggregated data (lines 12-13). For subsequent rounds or data instance, the client directly gets the previous aggregated data from the server without decryption (lines 14-16). In either case, it classifies the previous aggregated data using its local discriminator model (line 17). Then it masks the transformed data using the previous aggregated and transformed data (line 18). Finally, it sends the mask data to the server for data aggregation (lines 19-22).

In the server (i.e., Algorithm 2), the source cities act as the clients (line 1). During each training round and each traffic data (lines 2–3), if the training process is in the first round using the first data instance (line 4), the server gets the encrypted data from clients (line 5). Then, it aggregates them by summing up, and send the aggregated encrypted data to back to the clients for further processing (lines 6-7). For subsequent rounds or data instances (line 8), the server gets the mask data from clients (line 9). Then, it aggregates the masked data using the previous aggregated data (line 10). Next, it classifies the aggregated data using its global discriminator model and sends the aggregated data back to the clients (lines 11–14). Finally, at the end of each training round, it classifies local traffic data and performs traffic prediction using the aggregated and local traffic data (lines 15–17).

**Complexity Analysis**. We also give the complete complexity analysis for the training of the FedTT framework, i.e., Algorithms 1 and 2. For the client (i.e., Algorithm 1), the training complexity is $O((|\mathcal{M}^{R_i}| + |\mathcal{M}^S|) \times (F_1 \times H)^2 \times |\mathcal{X}^{R_i}|)$ at each round. For the server (i.e., Algorithm 2), the training complexity is $O((|\mathcal{M}^S| \times (F_1 \times H)^2 + MC(\theta_{TP})) \times (|\mathcal{X}^S| + \sum_{i=1}^{n} |\mathcal{X}^{R_i}|))$ at each round. Here, $|\mathcal{M}^{R_i}|$ and $|\mathcal{M}^S|$ are the number of sensors in the source city $R_i$ and target city $S$, respectively. Besides, $|\mathcal{X}^{R_i}|$ and $|\mathcal{X}^S|$ are the number of traffic data in the source city $R_i$ and target city $S$, respectively. In addition, $F_1 = 3$ is the dimensions of traffic data features, and $H = 1024$ is the hidden dimensions of the three-layer MLP model in $\theta_{Gen}^{R_i}$ and $\theta_{Dis}^{R_i}$. Moreover, $MC(\theta_{TP})$ is the model complexity of $\theta_{TP}$ (i.e., $\theta_{DyHSL}$).

## C.3   Theoretical Privacy Analysis

The privacy protection mechanism of the proposed FedTT framework comprises two stages. First, it uses the Traffic Domain Adapter (TDA) to transform the data from the traffic domain of source cities to that of the target city, where the parameters of the TDA model are private and not shared with the server and other clients. Second, it performs Traffic Secret Aggregation (TSA) to secure mask and aggregate the transformed data. Consequently, an attacker must first reverse-engineer the transformed data from the aggregated data and then infer the original traffic data from the transformed data. To rigorously analyze the privacy-preserving capability of these two stages, we first define the threat model as follows.

**Threat Model**. Following previous works [76, 60, 79] in federated learning scenarios, we assume that the server acts as a semi-honest adversary who will honestly execute the required operations (e.g., aggregation) but also remains curious about the private data in clients. In the FTT problem, the server may perform inference attacks to infer the raw instance-level traffic data of clients based on the adversary knowledge, including the client model architecture, privacy-preserving mechanism, and the intermediate data (e.g., model parameters or training gradients) uploaded by clients.

Based on this, we analyze the privacy leakage of FedTT using mutual information [30] as follows.

**Privacy Protection in Traffic Domain Adapter.** Given the transformed data $\mathcal{X}^{R_i \to S}$ of the source city $R_i$, the attacker aims to infer the original traffic data $\mathcal{X}^{R_i}$, where $\mathcal{X}^{R_i \to S}$ is derived from $\mathcal{X}^{R_i}$ in Eq.10 as shown below:

$$\mathcal{X}^{R_i} \xrightarrow{\theta_{TDA}} \mathcal{X}^{R_i \to S}, \tag{31}$$

where the TDA model $\theta_{TDA}$ is private and inaccessible. Since this process represents a deterministic mapping, the privacy leakage can be quantified as:

$$I(\mathcal{X}^{R_i}; \mathcal{X}^{R_i \to S}) = H(\mathcal{X}^{R_i \to S}) - H(\mathcal{X}^{R_i \to S}|\mathcal{X}^{R_i}) = H(\mathcal{X}^{R_i \to S}), \tag{32}$$

where $H(\cdot)$ denotes entropy and $H(\mathcal{X}^{R_i \to S}|\mathcal{X}^{R_i}) = 0$ due to the nature of deterministic mapping. Since $\mathcal{X}^{R_i \to S}$ is derived from $\mathcal{X}^{R_i}$ through the private TDA model $\theta_{TDA}$, the amount of privacy leakage can be further expressed as follows:

$$\begin{aligned}
I(\mathcal{X}^{R_i}; \mathcal{X}^{R_i \to S}) &\leq I(\mathcal{X}^{R_i}; \mathcal{X}^{R_i \to S}, \theta_{TDA}) \\
&= I(\mathcal{X}^{R_i}; \theta_{TDA}) + I(\mathcal{X}^{R_i}; \mathcal{X}^{R_i \to S}|\theta_{TDA}) \\
&= H(\mathcal{X}^{R_i \to S}|\theta_{TDA}) \propto \frac{|\mathcal{M}^{R_i}|}{|\theta_{TDA}| * |\mathcal{M}^S|},
\end{aligned} \tag{33}$$

where $|\theta_{TDA}|$ is the parameter space of the TDA model. As $\theta_{TDA}$ aligns the distribution of $\mathcal{X}^{R_i \to S})$ to the traffic domain of the target city through traffic domain alignment, reducing its correlation with the source city's traffic domain, $H(\mathcal{X}^{R_i \to S}|\theta_{TDA})$ takes on a small value, thereby minimizing the privacy leakage $I(\mathcal{X}^{R_i}, \mathcal{X}^{R_i \to S})$.

**Privacy Protection in Traffic Secure Aggregation.** Given the aggregated data $\overline{\mathcal{X}}^{\mathcal{R} \to S}$, the attacker aims to infer the transformer data $\mathcal{X}^{R_i \to S}$ of the source city $R_i$, where $\mathcal{X}^{(R_i \to S),R_i}$ is derived from $\mathcal{X}^{R_i \to S}$ in Eq.16 as shown below:

$$\overline{\mathcal{X}}^{\mathcal{R} \to S} = \frac{1}{n}(\mathcal{X}^{R_i \to S} + \sum_{j=1 \& j \neq i}^{n} \mathcal{X}^{R_j \to S}) \tag{34}$$

Since the traffic domains of source cities are aligned to that of the target city, they are from Independent Identically Distributed (IID), and the privacy leakage can be quantified as:

$$\begin{aligned}
I(\mathcal{X}^{R_i \to S}; \overline{\mathcal{X}}^{\mathcal{R} \to S}) &= H(\overline{\mathcal{X}}^{\mathcal{R} \to S}) - H(\overline{\mathcal{X}}^{\mathcal{R} \to S}|\mathcal{X}^{R_i \to S}) \\
&\leq H(\mathcal{X}^{R_i \to S}) - H(\frac{1}{n}\sum_{j=1 \& j \neq i}^{n} \mathcal{X}^{R_j \to S}) \\
&\leq \frac{H(\mathcal{X}^{R_i \to S})}{n} \propto \frac{1}{n * |\mathcal{M}^S|}
\end{aligned} \tag{35}$$

Since the above two processes is a Markov Chain [44], i.e., $\mathcal{X}^{R_i} \to \mathcal{X}^{R_i \to S} \to \overline{\mathcal{X}}^{\mathcal{R} \to S}$, the total amount of the privacy leakage can be bounded using the data processing inequality [56]:

$$\begin{aligned}
I(\mathcal{X}^{R_i}; \overline{\mathcal{X}}^{\mathcal{R} \to S}) &\leq min(I(\mathcal{X}^{R_i}; \mathcal{X}^{R_i \to S}), I(\mathcal{X}^{R_i \to S}; \overline{\mathcal{X}}^{\mathcal{R} \to S})) \\
&\leq min(H(\mathcal{X}^{R_i \to S}|\theta_{TDA}), \frac{H(\mathcal{X}^{R_i \to S})}{n})
\end{aligned} \tag{36}$$

This analysis demonstrates that the FedTT framework effectively minimizes privacy leakage by leveraging both TDA and TSA, ensuring robust privacy protection in federated traffic knowledge transfer.

# D   Experimental Details

## D.1   Baselines

We compare the FedTT framework with state-of-the-art baselines. First, we compare FedTT with three SOTA transfer methods in Federated Traffic Knowledge Transfer (FTT), including T-ISTGNN [49], pFedCTP [78], and 2MGTCN [75], as detailed below.

- **T-ISTGNN [49].** It designs a spatio-temporal GNN-based approach with an inductive mode for cross-region traffic prediction.

- **pFedCTP [78].** It designs an ST-Net for privacy-preserving and cross-city traffic prediction with personalized federated learning.

- **2MGTCN [75].** It designs multi-modal GCNs and TCNs to capture spatial and temporal information and enhance adaptability across cities.

Besides, we compare FedTT with three SOTA transfer methods in Multi-Source Traffic Knowledge Transfer (MTT), including TPB [41], ST-GFSL [43], and DastNet [57], as detailed below.

- **TPB [41].** It utilizes a traffic patch encoder to create a traffic pattern bank for the cross-city few-shot traffic knowledge transfer.
- **ST-GFSL [43].** It transfers traffic knowledge through model parameter matching to retrieve similar spatio-temporal features.
- **DastNet [57].** It employs graph learning and domain adaptation to create domain-invariant node embeddings for the traffic data.

In addition, we compare FedTT with three SOTA transfer methods in Single-Source Traffic Knowledge Transfer (STT), including CityTrans [47], TransGTR [27], and MGAT [46], as detailed below.

- **CityTrans [47].** It proposes a domain adversarial model with knowledge transfer for spatio-temporal prediction across cities.
- **TransGTR [27].** It leverages adaptive spatio-temporal knowledge and domain-invariant features for TP in data-scarce cities.
- **MGAT [46].** It extracts multi-granular regional features from source cities to enhance the effectiveness of knowledge transfer.

Moreover, we extend three classic and SOTA centralized traffic models in FedTT and the existing two-stage transfer methods in FTT (referred as FTL), including Gated Recurrent Unit (GRU) [10], Convolutional Neural Network (CNN) [32], Multi-Layer Perceptron (MLP) [55], CityTrans [47], TransGTR [27], and MGAT [46], as detailed below.

- **ST-SSL [23].** It models traffic data at attribute and structure levels for spatial and temporal heterogeneous-aware traffic prediction.
- **DyHSL [80].** It leverages hypergraph structure information to extract dynamic and high-order relations of traffic road networks.
- **PDFormer [24].** It introduces self-attention and feature transformation for dynamic and flow-delay-aware traffic prediction.

To evaluate the Traffic View Imputation (TVI) method of FedTT in the ablation study, we replace this module with three SOTA data imputation methods, including LATC [8], GCASTN [48], and Nuhuo [73], as detailed below.

- **LATC [8].** It integrates temporal variation as a regularization term to accurately impute missing spatio-temporal traffic data.
- **GCASTN [48].** It uses self-supervised learning and a missing-aware attention mechanism to impute the missing traffic data.
- **Nuhuo [73].** It uses graph neural networks and self-supervised learning to accurately estimate missing traffic speed histograms.

## D.2 Model Adaptability

Table 4 shows the overall performance when extending existing centralized traffic models (i.e., GRU [10], CNN [32], MLP [55], CityTrans [47], TransGTR [27], and MGAT [46]) in FTT using FedTT and FTL methods with MAE, where the best results are shown in blue. As observed, all centralized traffic models extended in FedTT achieve the best performance compared to those extended in FTL, also showing its effectiveness of traffic knowledge transfer in FTT, i.e., the gains range from **5.13% to 64.65%**. Note that the DyHSL model has the best performance in centralized traffic models and is implemented in FedTT as the default model in other experiments.

## D.3 Training Efficiency

Table 4: The overall performance (MAE) comparison when extending centralized traffic models

| Model | Method | (P8, FT, HK) → P4 | | | (P4, FT, HK) → P8 | | | (P4, P8, HK) → FT | | | (P4, P8, FT)→ HK | | |
|---|---|---|---|---|---|---|---|---|---|---|---|---|---|
| | | flow | speed | occ | flow | speed | occ | flow | speed | occ | flow | speed | occ |
| GRU | FTL[1] | 29.27 | 3.39 | 0.0282 | 23.44 | 2.40 | 0.0253 | 21.16 | 12.18 | 0.0712 | 10.11 | 4.60 | 0.0125 |
| | FedTT | 25.93 | 2.24 | 0.0220 | 20.73 | 2.21 | 0.0213 | 17.34 | 5.67 | 0.0401 | 9.33 | 2.86 | 0.0101 |
| CNN | FTL | 31.46 | 4.55 | 0.0317 | 27.60 | 3.27 | 0.0267 | 24.55 | 9.05 | 0.0803 | 9.74 | 5.92 | 0.0169 |
| | FedTT | 26.82 | 2.84 | 0.0274 | 22.20 | 2.41 | 0.0217 | 17.44 | 6.27 | 0.0472 | 9.24 | 3.92 | 0.0113 |
| MLP | FTL | 34.01 | 3.66 | 0.0276 | 30.24 | 2.88 | 0.0246 | 22.66 | 14.43 | 0.0743 | 10.87 | 5.23 | 0.0146 |
| | FedTT | 28.08 | 2.17 | 0.0250 | 23.79 | 2.40 | 0.0212 | 17.66 | 7.35 | 0.0480 | 9.68 | 3.27 | 0.0102 |
| ST-SSL | FTL | 26.76 | 2.26 | 0.0176 | 20.06 | 1.88 | 0.0226 | 19.43 | 7.78 | 0.0605 | 9.43 | 4.36 | 0.0117 |
| | FedTT | 22.28 | 1.34 | 0.0096 | 17.14 | 1.27 | 0.0114 | 13.38 | 4.88 | 0.0400 | 8.76 | 1.65 | 0.0097 |
| DyHSL | FTL | 18.61 | 1.39 | 0.0131 | 16.71 | 1.40 | 0.0144 | 16.96 | 6.04 | 0.0324 | 8.63 | 2.97 | 0.0103 |
| | FedTT | 16.69 | 1.03 | 0.0061 | 14.11 | 0.94 | 0.0059 | 12.10 | 3.24 | 0.0249 | 7.42 | 1.05 | 0.0087 |
| PDFormer | FTL | 26.99 | 2.31 | 0.0194 | 22.85 | 1.80 | 0.0232 | 17.92 | 6.57 | 0.0433 | 9.17 | 3.29 | 0.0108 |
| | FedTT | 22.05 | 1.43 | 0.0125 | 17.67 | 1.36 | 0.0127 | 13.09 | 3.53 | 0.0314 | 8.22 | 1.22 | 0.0091 |

[1] FTL refers to the two-stage method of existing methods in FTT.

Fig. 5 shows the communication size (GB) and running time (minutes) of different methods on traffic flow prediction. As observed, the FedTT framework has the least communication size and running time compared to other methods, i.e., with communication overhead reduced by **90%** and running time reduced by **1 to 2 orders of magnitude**, showing its superior efficiency of traffic knowledge transfer in FTT. This is because FedTT securely transmits and aggregates the

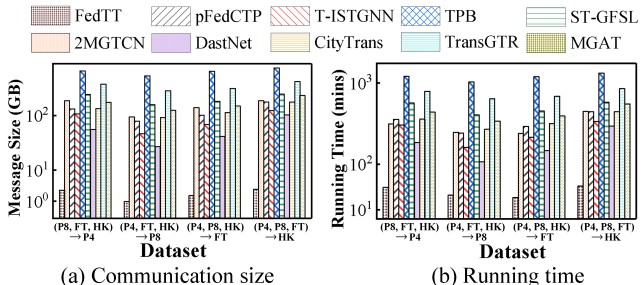

(a) Communication size     (b) Running time

Table 5: Training efficiency study of different methods

traffic domain-transformed data using the TST module with relatively small computation and communication overheads, compared to other methods that employ the HE method for model secure aggregation in FTT. Besides, FedTT utilizes the FPT module to reduce data transmission and train models in parallel, significantly improving the training efficiency in FTT.

### D.4 Parameter Sensitivity

Fig. 6 shows the performance of the FedTT framework with different hyperparameter settings (i.e., $\lambda_1$ and $\lambda_2$) on traffic flow prediction with MAE. First, the suggestion and optimum value of $\lambda_1$ is 0.7. As $\lambda_1$ increases, the generator model tends to generate the data that can "trick" the server discriminator model rather than generating the high-quality traffic domain transformed data, resulting in higher MAE. As $\lambda_1$ decreases, the server dis-

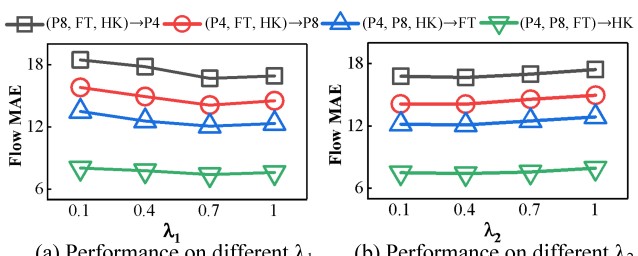

(a) Performance on different $\lambda_1$     (b) Performance on different $\lambda_2$

Table 6: Parameter sensitivity of FedTT

criminator model loses its ability to effectively guide the generator model in generating traffic domain transformed data, resulting in higher MAE. Second, the suggestion and optimum value of $\lambda_2$ is 0.4. As $\lambda_2$ increases, the generator model tends to generate the data with a traffic domain that deviates significantly from that of the target city, resulting in higher MAE. As $\lambda_2$ decreases, the generator model generates the data with a more local-specific traffic pattern, which hinders the model from effectively learning the traffic patterns of the target city, resulting in higher MAE. Overall, FedTT has the best performance in all hyperparameter settings when $\lambda_1 = 0.7$ and $\lambda_2 = 0.4$, which are used in FedTT as the default values in other experiments.

### D.5 Case Study

To demonstrate the practical applicability of FedTT in real-world traffic knowledge transfer scenarios, we conduct a case study using the UTD19[42] dataset, which includes traffic data from 40 cities

Table 7: Statistics of evaluated cities in UTD19

| City | # instances | # sensors | Interval | Missing Rate |
|---|---|---|---|---|
| London | 6454 | 5719 | 5 min | 19.47% |
| Hamburg | 50142 | 418 | 3 min | 2.66% |
| Manchester | 6984 | 181 | 5 min | 10.61% |
| Madrid | 4560 | 1116 | 5 min | 16.02% |
| Groningen | 525 | 55 | 5 min | 1.75% |

worldwide. For comparison, we select 2MGTCN, as it performs the best among the three existing methods in FTT (see Table 4.1). In this scenario, Groningen is chosen as the target city due to its limited traffic data and relatively sparse sensor deployment, making it challenging to train a high-performance traffic model independently. In contrast, London, Hamburg, Madrid, and Manchester are chosen as source cities because they possess significantly larger datasets and denser sensor networks, providing abundant traffic data for effective knowledge transfer. The statistics of these cities is summarized in Table 7. Since the sampling intervals of traffic data vary across cities, we resample all datasets in a uniform interval of 15 minutes to ensure that the temporal discrepancies between cities do not affect the model performance.

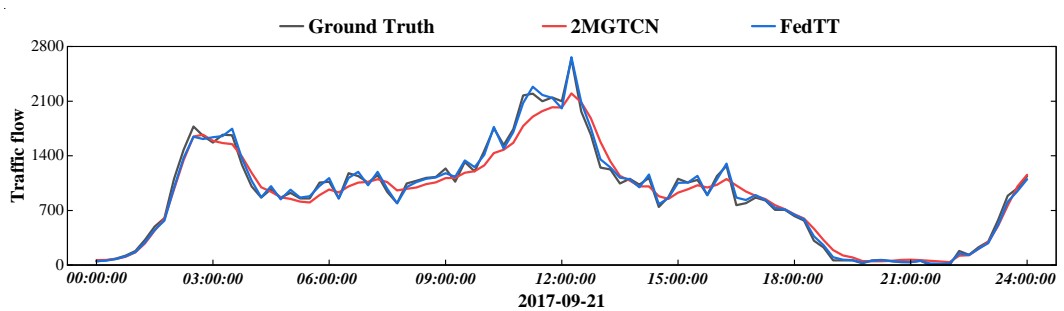

(a) Sensor PGR01_101725_G172_Emmaviaduct_Z_ID_8650_1

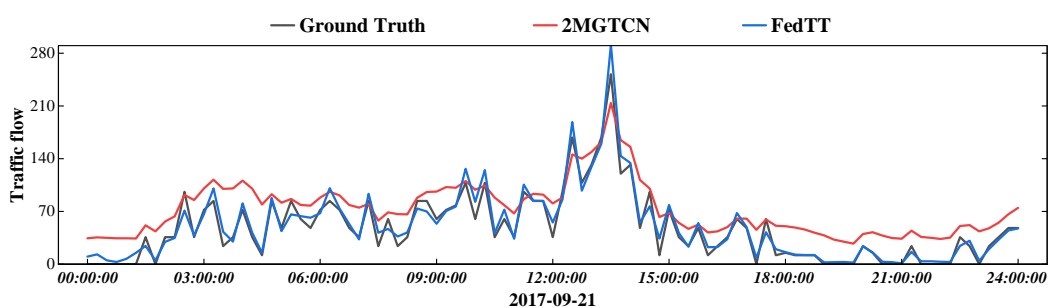

(b) Sensor PGR01_101727_Hereweg_Z_ID_8610_2

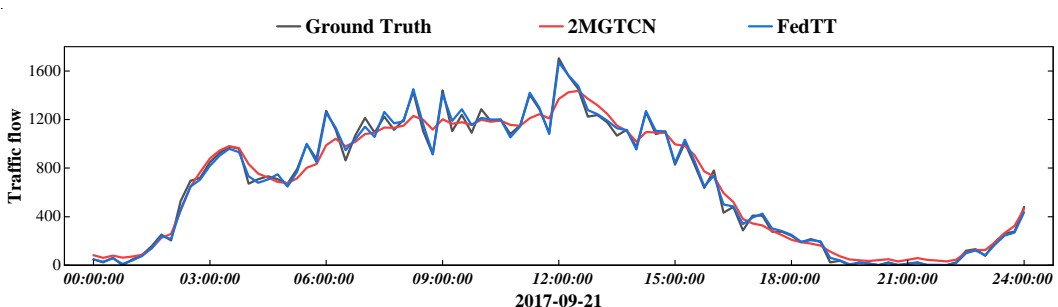

(c) Sensor PGR01_101761_Sontweg_NO_ID_8812_1

Figure 10: Visualization of traffic flow prediction in groningen

The traffic flow results of three sensors (i.e., PGR01_101725_G172_Emmaviaduct_Z_ID_8650_1, PGR01_101727_Hereweg_Z_ID_8610_2, and PGR01_101761_Sontweg_NO_ID_8812_1) on September 21, 2017 in Groningen are shown in Fig. 10. As observed, the prediction of FedTT aligns well with the ground truth, while 2MGTCN can only learn the general trend of traffic flow. Taking sensor PGR01_101761_Sontweg_NO_ID_8812_1 as an example. FedTT and 2MGTCN excels from 0:00 a.m. to 6:00 a.m., a period characterized by relatively smooth traffic flow. Through-out the peak hours, from 6 a.m. to 6 p.m., when traffic flow fluctuations are pronounced, FedTT showcases adaptability by learning from the rapid increase and decrease in traffic, while 2MGTCN predicts a relatively smooth traffic flow that does not match the real one. Between 6 p.m. and 12 a.m., as the traffic flow gradually decreases and stabilizes, FedTT maintains relatively accurate predictions compared to 2MGTCN. In summary, the FedTT framework demonstrates its robust performance on real-world traffic knowledge transfer scenarios, yielding satisfactory and accurate prediction results in forecasting the traffic flow across different periods.

