# OpenReview forum: "Privacy-Preserving and Effective Cross-City Traffic Knowledge Transfer via Federated Learning"
_NeurIPS.cc/2025/Conference — Submitted to NeurIPS 2025_

### Official Review · Reviewer_jrXL · 2025-06-03

**Clarity:** 2
**Significance:** 3
**Originality:** 1
**Rating:** 2
**Confidence:** 4

**Summary:**

The manuscript titled "Privacy-Preserving and Effective Cross-City Traffic Knowledge Transfer via Federated Learning" proposes FedTT, a federated learning framework aimed at enabling privacy-preserving knowledge transfer for city-level traffic prediction tasks. While the application of federated learning in traffic modeling is timely and relevant, the manuscript presents several critical shortcomings in novelty.

**Questions:**

N/A

**Ethical Concerns:**

["NO or VERY MINOR ethics concerns only"]

**Limitations:**

The manuscript does not sufficiently discuss the practical limitations of the proposed approach in real-world deployments. Can the authors elaborate on scenarios where FedTT may not perform well or where its applicability is restricted? Understanding these constraints is critical for assessing its utility in operational systems

**Quality:**

1

**Strengths And Weaknesses:**

1. The proposed approach lacks sufficient novelty. Similar frameworks and strategies have already been explored in recent literature within this domain.
2. The introduction section does not adequately discuss the limitations of existing methods, nor does it clearly position how the proposed work addresses these gaps.
3. Background information necessary to contextualize the work is missing or insufficient, making it difficult to assess the contribution in relation to prior art.
4. The methodology is presented at a high level without sufficient technical depth or detailed explanations of critical components.
5. The manuscript suffers from poor structural organization and weak presentation, which hinders readability and understanding.
6. The evaluation uses a narrow set of metrics, which does not fully capture the performance or robustness of the proposed method.
7. Visual elements, including figures, are poorly formatted and lack clarity—font sizes are small, structures are imprecise, and the visualizations are not sufficiently informative.

---

> ### Author Rebuttal · Authors · 2025-07-31
>
> Thanks for the comments and questions.
> ```
> W1. The proposed approach lacks sufficient novelty. Similar frameworks and strategies have already been explored in recent literature within this domain.
> ```
> **We sincerely appreciate the opportunity to elaborate on FedTT's novel contributions**. Our framework introduces several key innovations that, to our knowledge, represent the first comprehensive solution addressing three fundamental challenges in Federated Traffic Knowledge Transfer (FTT): **privacy leakage risk, data distribution discrepancies, and low data quality**.
>
> To address these challenges, FedTT features three carefully designed modules: Traffic View Imputation (**TVI**), Traffic Domain Adapter (**TDA**), and Traffic Secret Aggregation (**TSA**), **which all are different from existing literature/methods**.
>
> + **To address low data quality**, the TVI module enhances the completeness and reliability of traffic data by imputing missing values through comprehensive modeling of spatial and temporal dependencies. While existing imputation techniques exist, they often fail to fully exploit the complex correlations inherent in traffic data, leading to suboptimal performance.
> + **To address data distribution discrepancies**, the TDA module performs domain adaptation by transforming source data into the target domain. While multi-source traffic knowledge transfer also focuses on reducing data distribution discrepancies, it relies on centralized frameworks, which involve sharing and exchanging traffic data across cities without considering traffic data privacy.
> + **To address privacy leakage risks**, the TSA module securely aggregates domain-adapted representations across clients. Unlike homomorphic encryption and differential privacy, which introduce high overhead or degrade model utility, TSA is a lightweight solution that preserves model performance without compromising privacy or efficiency.
> + **Extensive experiments** across diverse settings demonstrate the effectiveness and robustness of FedTT, which significantly outperforms 14 strong baselines across multiple categories with MAE reduction by 5.43% to 75.24%.
>
> ```
> W2. The introduction section does not adequately discuss the limitations of existing methods, nor does it clearly position how the proposed work addresses these gaps.
> ```
> We thank the reviewer for this valuable feedback. In the current version, **we have indeed discussed the limitations of existing methods** (Page 2), namely: (1) **privacy leakage risk**, (2) **data distribution discrepancies**, and (3) **low data quality**, all of which are well-known and non-trivial challenges in this domain. To address these issues, we propose **FedTT**, a novel and unified framework specifically designed to tackle these three challenges through three dedicated modules: **TVI** (for data imputation), **TDA** (for domain adaptation), and **TSA** (for secure aggregation).
>
> ```
> W3. Background information necessary to contextualize the work is missing or insufficient, making it difficult to assess the contribution in relation to prior art.
> ```
> In the current version, **we provide essential background to contextualize our work**. Specifically,
>
> + **In the Introduction Section**, we clearly establish the context by describing the background and motivation. Then, we explicitly identify three key challenges that limit the application of this existing framework in real-world scenarios.
> + **The Problem Definition Section** provides the necessary mathematical and conceptual foundation by formally defining core concepts such as road networks, traffic data, and the FTT problem.
> + **In Appendix A** (page 15), we provide a comprehensive review of the related work, which is organized into traffic prediction and traffic knowledge transfer. Specifically, we discuss recent advancements in traffic prediction and review existing methods in single-source, multi-source, and federated traffic knowledge transfer.
>
> ```
> W4. The methodology is presented at a high level without sufficient technical depth or detailed explanations of critical components.
> ```
> We thank the reviewer for the comment. **We would like to clarify that our methodology is presented with substantial technical depth and detailed descriptions of all critical components**, as also recognized by all the other reviewers. Specifically, the methodology section is organized in a modular and systematic manner to facilitate clarity and depth:
>
> + **Method design motivations** are explicitly provided for each core module: TVI, TDA, and TSA, to explain the specific challenges each component is designed to address. Then, each module is accompanied by a **dedicated subsection** (Sections 3.1–3.3), where we formally define the operations and workflow.
> + We include **extensive mathematical formulations (Eqs. 6--21)** to rigorously specify the algorithms. For instance, the adversarial training losses in TDA  are detailed in Eqs. 14, 15, and 20, and the privacy-preserving masking mechanism in TSA is defined in Eq. 16.
> + A **complete training algorithm** is also provided (Algorithm 1, page 17), outlining the step-by-step execution of FedTT on both the client and server sides.
> + **All the code, datasets, and running instructions are provided for reproducibility**. These materials include step-by-step instructions for reproducing our experimental results.
>
> We believe that the combination of formal definitions, mathematical rigor, and a full algorithmic overview provides the necessary technical depth to fully understand and implement FedTT.
>
> ```
> W5. The manuscript suffers from poor structural organization and weak presentation, which hinders readability and understanding.
> ```
> The paper organization follows a standard and coherent flow:
> + **Section 1** (Introduction) establishes the context, identifies three key challenges in existing work, and motivates our approach.
> + **Section 2** (Problem Definition) formally defines the core concepts and the FTT problem.
> + **Section 3** (Methodology) is the core of the paper, which is divided into three dedicated subsections to present our three key modules, i.e., TVI, TDA, and TSA, each with a clear design motivation, technical details, and mathematical formulations.
> + **Section 4** (Experiments) is then structured to first present the experimental setup, followed by results that directly address the three challenges: privacy protection (Fig. 6), effectiveness (Fig. 7, Table 2), and robustness (Figs. 8 and 9).
> + **Section 5** concludes the paper and discusses limitations.
>
> ```
> W6. The evaluation uses a narrow set of metrics, which does not fully capture the performance or robustness of the proposed method.
> ```
> Thanks for pointing this out. In the current version, **we have employed all standard metrics that are consistent with prior work in this field [1–3]** and designed to assess the three core aspects of the proposed FedTT framework:
>
> - **Effectiveness**: We report task-specific performance metrics (e.g., MAE, RMSE) to measure the accuracy of the final predictions.
> - **Efficiency**: We evaluate communication and computational overhead to assess the training cost and practical applicability of FedTT in real-world scenarios.
> - **Privacy Protection**: We conduct both theoretical analysis (Appendix C.3) and empirical evaluations (Section 4.2) to demonstrate that FedTT maintains strong privacy guarantees without sacrificing model utility.
>
> We would like to emphasize that, the metrics used in our experiments already exceed the metrics in comparable baselines. Also, **we're happy to incorporate additional metrics if the review could specify which ones would be most relevant**.
>
> *[1] Federated Transfer Learning for Privacy-Preserved Cross-City Traffic Flow Prediction. TITS, 2025.*
>
> *[2] Personalized Federated Learning for Cross-City Traffic Prediction. IJCAI, 2024.*
>
> *[3] Privacy-Preserving Cross-Area Traffic Forecasting in ITS: A Transferable Spatial-Temporal Graph Neural Network Approach. TITS, 2023.*
>
> ```
> W7. Visual elements, including figures, are poorly formatted and lack clarity—font sizes are small, structures are imprecise, and the visualizations are not sufficiently informative.
> ```
> Thanks for the suggestion. In response, **we will revise all visual elements** to **ensure higher resolution, larger fonts, and clearer graphical structures**.
>
> ```
> Limitations. The manuscript does not sufficiently discuss the practical limitations of the proposed approach in real-world deployments. Can the authors elaborate on scenarios where FedTT may not perform well or where its applicability is restricted? Understanding these constraints is critical for assessing its utility in operational systems.
> ```
> We thank the reviewer for this insight. Actually, **we have indeed discussed the limitations**, titled "Conclusion and Limitations".
>
> - **Urban heterogeneity**: Our current evaluation focuses on city-level scenarios with standard road networks. FedTT has not yet been tested on more complex grid-based or irregular urban layouts, where spatial correlations may differ significantly. This represents a valuable direction for future research.
> - **Task generalization**: While our framework is designed and evaluated primarily for traffic prediction, its effectiveness in other spatio-temporal tasks (e.g., crowd flow prediction, event detection) remains to be validated.
> - **Scalability**: We have not systematically analyzed the impact of increasing the number of participating source cities. But, during this rebuttal phase, we have conducted additional experiments by varying the number of source cities (clients) to 4, 8, 12, and 16, respectively. The results can be found in **Response to Reviewer imHF.W2**, which indicate that FedTT consistently maintains robust performance and effective knowledge transfer across varying numbers of source cities with **MAE reduction by 17.13% to 39.61%**, demonstrating its scalability and stability.

---

> ### Author Response · Authors · 2025-08-06
> **Kind Reminder: Request for Discussion on Our Rebuttal**
>
> Dear Reviewer jrXL,
>
> Thank you again for your time with our manuscript. We truly appreciate your insights and the efforts you’ve put into evaluating our work.
>
> We’ve had the opportunity to address all the other reviewers’ concerns in our rebuttal, and we’re grateful that the other reviewers have acknowledged our contributions and found our responses satisfactory. **However, as we highly value your perspective, we would be especially grateful if you could share your thoughts on whether our rebuttal has adequately addressed your concerns**.
>
> **As the discussion period is nearing its end**, please don’t hesitate to let us know if you need any clarifications from our side. Thank you once more for your contributions to the review process.
>
> Best regards,
>
> The Authors of Paper 7672

---

### Official Review · Reviewer_jzay · 2025-06-08

**Clarity:** 3
**Significance:** 2
**Originality:** 2
**Rating:** 3
**Confidence:** 3

**Summary:**

This paper proposes FedTT, a framework designed for cross-city traffic knowledge transfer with privacy preservation. The paper introduces three key components: (i) TVI: missing traffic data in each city is completed using GAT and DyHSL; (ii) TDA: a GAN transforms source city data into the style of the target city to generate synthetic data; and (iii) TSA: synthetic data generated from source cities are transmitted to the target city and then averaged. Experiments are conducted on four real-world traffic datasets, comparing FedTT with many SOTA methods, along with ablation studies. The results show that FedTT outperforms these baselines in different aspects.

**Questions:**

1. Please refer to the Weaknesses section and clarify how FedTT differs from other one-shot FL frameworks.

2. I believe the paper should only be compared with methods that also transmit data, rather than model parameters. This is because comparing fundamentally different paradigms is unfair. Each comes with its own pros and cons. A major limitation of FedTT is that it does not yield a reusable global model; the trained model cannot be generalized to an unseen city. When facing a new city, FedTT would require re-collecting data and retraining the model from scratch.

3. The necessity of TSA is questionable as it is executed on the server side. In FL, it is commonly assumed that the server is honest but cautious. The server could ignore the aggregation protocol and easily infer client data from Eq. (16). Furthermore, aggregation can reduce data volume, diversity, and city-specific granularity.

4. The training procedure of the GAN is unclear. If communication between clients and the server is limited to data transfer, is the discriminator on the server the same model as the one used on the client side, or is it merely the same architecture trained independently? If the generator is trained on the client and the discriminator on the server, how is this adversarial training coordinated?

5. FedTT assumes that the server (i.e., the target city) has access to 5% of its own training data and computation resources. Do all baseline methods assume access to the same amount of target city data and computational resources as FedTT?

**Ethical Concerns:**

["NO or VERY MINOR ethics concerns only"]

**Final Justification:**

I agree with most of the authors' responses.
Q1. I think there was a previous misunderstanding about the training and communication in the paper.
Q2. I reached consensus with the authors that the proposed FedTT cannot adapt to new cities, which is a limitation.
Q3. I agree with the authors' response.
Q4. The authors' response resolved my confusion, but such a training method may be challenging, especially when the number of clients increases. This could lead to gradient concerns or more effort required for hyperparameter tuning.
Q5. I cannot fully accept the authors' response. The proposed FedTT requires the server (target city) to have training data and computational resources, while for the baseline methods, although some use fine-tuning, it is not mandatory.

Besides this, I think the proposed FedTT might be a supplement to the traffic knowledge transfer field, as I also observed that all baseline methods compared in the paper transfer models rather than data. Setting aside the reasonableness of transferring models vs. data, such a method might be new in the traffic knowledge transfer field. However, transferring data is an existing approach in the broader FL problem. Therefore, in my opinion, I believe the proposed method does not make significant contributions to FL, because the proposed method is not a generalized approach, and there are also some additional prerequisites including server resources, as well as deficiencies in data transfer. So, I also cannot confidently assert that it makes very significant contributions to the traffic knowledge transfer field.

**Limitations:**

FedTT lacks generalization to unseen cities.

**Quality:**

3

**Strengths And Weaknesses:**

Strengths:
The paper is clearly written and well organized. One of its main strengths is the use of real-world traffic datasets. It considers a lot of SOTA methods, and the experiments are comprehensive, not only focusing on prediction error but also considering communication overhead, runtime, and privacy protection.

Weaknesses (revised after rebuttal):
The clarity of the paper writing needs improvement, including the missing training processes for generators and discriminators in Algorithms 1 and 2, parameter settings such as the configured training rounds and corresponding communication rounds, as well as loss weights. How to ensure stable training of the generators and discriminators on clients and the discriminator on the server is also a challenge, especially when the number of clients increases.

---

> ### Author Rebuttal · Authors · 2025-07-31
>
> Thanks for all the valuable comments and questions. **We have addressed all the concerns as detailed below**.
> ```
> W1. The novelty is weak.
> ```
> **Thanks for the opportunity to clarify the novelty and contributions of our work, which have been positively acknowledged by other reviewers**. Existing approaches to Federated Traffic Knowledge Transfer (FTT) face three non-trivial challenges: **(1) privacy leakage risk, (2) data distribution discrepancies, and (3) low data quality**.
>
> To the best of our knowledge, no prior work addresses these issues in a unified federated setting. Our proposed framework, **FedTT**, introduces three key and novel modules: **Traffic View Imputation (TVI), Traffic Domain Adapter (TDA), and Traffic Secret Aggregation (TSA)**, which are explicitly designed to address these challenges.
> + TVI for Low Data Quality. We propose the TVI module to enhance traffic data completeness by capturing complex spatial and temporal dependencies. **Compared to prior techniques that either ignore missing data or apply simplistic imputation**, TVI enables FedTT to operate effectively in real-world and low-quality data scenarios.
> + TDA for Cross-City Distribution Discrepancies. The TDA module performs adversarial domain alignment by transforming source city data distributions to match that of the target city. While some existing methods also aim to reduce distribution shift, **they typically rely on centralized architectures that require direct data sharing**, which is prohibited in FL.
> + TSA for Privacy-Preserving Aggregation. To safeguard data privacy, we design the TSA module to securely transmit and aggregate the transformed data across cities. **Unlike homomorphic encryption and differential privacy, which introduce high overhead or degrade model utility**, TSA is a lightweight solution without compromising privacy or accuracy.
> + Extensive experiments validating both its effectiveness and practicality that **FedTT outperforms 14 strong baselines with MAE reduction by 5.43% to 75.24%**. In summary, **FedTT is a novel solution** that jointly addresses three critical challenges in FTT, presenting a significant advancement over existing methods.
> ```
> W2&Q1. I believe FedTT is essentially a one-shot federated learning. Please clarify the difference. The paper does not compare with other data-transmitting FL methods like one-shot FL.
> ```
> **We argue that FedTT does not fall under the category of one-shot federated learning**, and there are fundamental differences between FedTT and one-shot FL.
> - One-shot FL [1--2] typically refers to a paradigm where 1) the server sends an initial global model to clients, 2) each client conducts local training on its dataset, and 3) clients return the trained local models for a **single** aggregation. Its core goal is to **minimize communication overhead**, often at the cost of reduced model accuracy and adaptability. Furthermore, such frameworks usually **transmit model parameters instead of data**.
> - In contrast, **FedTT is an iterative multi-round federated transfer learning framework** specifically designed to address three critical challenges in the traffic knowledge transfer setting. Before the main training, each client first performs imputation via TVI. During FedTT's training, source data is transformed to the target domain via TDA. Then, the transformed data is securely transmitted and aggregated via TSA. Finally, the aggregated data are employed to train the target traffic model. This process is **iterative**, involving repeated client-server interaction over multiple rounds, **which is in stark contrast to the single-round communication of one-shot FL**.
>
> *[1] Revisiting Ensembling in One-Shot Federated Learning. NeurIPS, 2024.*
>
> *[2] FedLPA: One-shot Federated Learning with Layer-Wise Posterior Aggregation. NeurIPS, 2024.*
> ```
> W3&Q2. All baselines transmit model parameters, while FedTT transmits data, which is an unfair comparison.
> ```
> **We respectfully disagree with the assertion that our comparison is unfair**.
> - **The claim that FedTT "transmits data" while other methods "transmit model parameters" oversimplifies the nature of information exchange in federated learning**. In reality, **all FL frameworks transmit data-derived information**, whether in the form of model updates (e.g., gradients, parameters) or intermediate representations (e.g., transformed data, **domain-adapted latent representations** in FedTT). Therefore, the distinction between "data" and "model" transmission is not a meaningful basis for evaluating fairness in FL comparisons.
> - Furthermore, our evaluation is comprehensive, as we compare FedTT against all relevant categories in this field: 1) federated traffic knowledge transfer methods, 2) multi-source traffic knowledge transfer methods, and 3) single-source traffic knowledge transfer methods. **The baseline selection criteria and experimental setup align with the prior work [3--4] to ensure a fair evaluation**.
>
> *[3] Federated Transfer Learning for Privacy-Preserved Cross-City Traffic Flow Prediction. TITS, 2025.*
>
> *[4] Personalized Federated Learning for Cross-City Traffic Prediction. IJCAI, 2024.*
> ```
> Q2&Limitation. FedTT lacks generalization to unseen cities, requiring retraining from scratch for each new city.
> ```
> We sincerely thank the reviewer for this critical observation. We acknowledge that the traffic model trained for a specific target city in FedTT cannot be directly applied to unseen cities.
> + This limitation arises from the dual nature of urban traffic patterns, which consist of: **global patterns** (e.g., daily/weekly periodicity) that are common across cities, and **local patterns** (e.g., road topology, traffic regulations) that are unique to an individual city.
> + **FedTT is specifically designed to capture both traffic patterns for maximum prediction effectiveness**. It learns not only the global pattern from the source cities, but also the local pattern of the target city by transforming source data to the target domain via TDA. This specialization enables FedTT's SOTA performance for the target city (as shown in our experiments), whereas **conventional FTT methods that focus solely on global patterns sacrifice accuracy for broader applicability**.
>
> We appreciate the reviewer for highlighting this **trade-off between model specificity and generalizability**. This is a valuable supplement, and we will discuss this point in the next manuscript.
> ```
> Q3. (1) The necessity of TSA is questionable as it is executed on the server, which could ignore the aggregation and infer client data via Eq. 16. (2) Aggregation can reduce data volume, diversity, and city-specific granularity.
> ```
> We appreciate the reviewer's concerns regarding the necessity of the **Traffic Secret Aggregation (TSA)** module. **We would like to clarify a possible misunderstanding regarding how TSA enforces privacy protection**.
> + TSA is a **client-server protocol**, and its core privacy-preserving mechanism lies in the masking operation executed on the client before communication. As detailed in Eq. (16) and Algorithm 1 (page 18), each client masks its transformed data using the aggregated data from the last communication round. This ensures that the data sent to the server is already obfuscated on the client side.
> + Upon receiving the masked data, the server performs aggregation (Eqs. 17--18), and **does not have access to the unmasked data of any individual client**. Because the masking is based on shared historical aggregated statistics, the server cannot invert the process or isolate individual client contributions, under an honest-but-curious assumption. This provides a **strong privacy guarantee**, which we have validated through both **empirical experiments (Section 4.2) and theoretical privacy analysis (Appendix C.3, page 19)**.
> + Regarding the concern that aggregation might reduce data diversity or dilute city-specific signals: Our design choice is intentional. Aggregating domain-aligned representations from source cities allows us to **mitigate overfitting to any one city's idiosyncrasies** and encourages the model to **better adapt to the target city's traffic patterns**.
> ```
> Q4. Clarify GAN training: Are client and server discriminators the same model or independently trained? How is adversarial training coordinated if the client trains the generator and the server trains the discriminator?
> ```
> We appreciate the opportunity to clarify the adversarial procedure in FedTT. Actually, **both the client and server have their own discriminator models**, which are trained collaboratively in the adversarial process. While we implemented MLPs for both, **their architectures can be different**, as their roles are distinct. The training of adversarial components is coordinated.
> + On the client, **the generator** $\theta^{R_i}\_\textit{Gen}$ transforms the local source data into the target domain. Meanwhile, **the local discriminator** $\theta^{R_i}\_\textit{Dis}$ is trained to distinguish between local and aggregated transformed data. This encourages the generator to produce data that is consistent with the aggregated data, reducing the influence of source cities' local patterns.
> + On the server, **the global discriminator** $\theta_\textit{Dis}$ is trained to distinguish the aggregated data from its local target data. This encourages the generator to produce data that is aligned with the local traffic patterns of the target city.
>
> The coordination is achieved via the generator's loss (Eq. 21), which incorporates feedback from both discriminators, creating a unified adversarial process. The generator aims to fool both, while discriminators are trained to classify correctly via Eqs. 14 and 20.
> ```
> Q5. Do baselines assume access to the same amount of target city data and computational resources as FedTT?
> ```
> Yes, **all baselines follow the same setting for a fair comparison**.

---

> > ### Comment · Reviewer_jzay · 2025-08-03
> >
> > Thank you for your detailed response. I still have several confusions that I hope to get further clarification on:
> >
> > **Q1:** I previously mistakenly thought that FedTT only had one communication round. The reason for my misunderstanding is that Algorithm 1 and Algorithm 2 might have some confusion. Does the training round = communication round? I cannot find the specific value of training rounds used in the experiments. If FedTT and the baseline methods have the same number of communication rounds, how does it achieve very small communication size as shown in Table 5(a)? Algorithm 1 and Algorithm 2 also don't mention the training of generators and discriminators, which leads to the confusion in **Q4**. Could you include the training procedure of the generator and discriminators in Algorithm 1 and Algorithm 2?
> >
> > **Q2 & Q5:** I think transmitting data vs. transmitting models is one **taxonomy** of FL, and there are obviously other taxonomies here, such as the more fine-grained one mentioned by the authors. My view is that, generally speaking, they have different advantages and disadvantages. For example, FedTT requires the target city to have training data and substantial computational resources, while baseline methods may not need them, only requiring aggregation computation, which requires far fewer resources than training. Moreover, model-transmitting FL can more easily scale to new cities. In addition, privacy attack methods are definitely different for transmitting data vs. models, whether dealing with processed raw data, synthetic data, intermediate representations, or gradients and parameters.
> >
> > **Q3:** I agree with you.

---

> > > ### Author Response · Authors · 2025-08-03
> > > **Further Clarification for the Confusions Raised by Reviewer jzay**
> > >
> > > Thank you for the follow-up questions. We sincerely appreciate your careful reading of our work and your willingness to engage in this discussion. **We believe that we have addressed all the follow-up concerns**.
> > > ```
> > > Q1. I previously mistakenly thought that FedTT only had one communication round. The reason for my misunderstanding is that Algorithm 1 and Algorithm 2 might have some confusion. Does the training round = communication round? I cannot find the specific value of training rounds used in the experiments. If FedTT and the baseline methods have the same number of communication rounds, how does it achieve very small communication size as shown in Table 5(a)? Algorithm 1 and Algorithm 2 also don't mention the training of generators and discriminators, which leads to the confusion in Q4. Could you include the training procedure of the generator and discriminators in Algorithm 1 and Algorithm 2?
> > > ```
> > > Thank you for your interest and valuable questions regarding the training rounds, communication efficiency, and the training of generators and discriminators in **FedTT**. We address each concern as follows:
> > >
> > > ### **1. Training Rounds vs Communication Rounds**
> > >
> > > In FedTT, **training rounds are not equivalent to communication rounds**.
> > > - We adopt **1000 training rounds** in our experiments. Due to space limitation, this detail is currently available in our code repository. But we will explicitly clarify it in the revised version of the paper to improve clarity.
> > > - Communication is **not performed in every training round**. Instead, we implement a **Federated Parallel Training** strategy (Appendix C.1, Page 17) that is one of our contributions, where only certain modules are updated and communicated **every 5 training rounds**. This significantly reduces the total number of communication rounds to **200**, thus reducing the frequency of communication and improving training efficiency.
> > >
> > > ### **2. Communication Efficiency of FedTT (Table 5a)**
> > >
> > > FedTT achieves a **significantly lower communication cost** compared to baseline methods due to two key design choices:
> > > - **Lightweight data transmission**:
> > >    Unlike baselines that transmit high-dimensional model parameters or gradients (often millions of floats), FedTT transmits only **domain-transformed traffic data**. The size of this data is determined by the number of sensors in the target city (e.g., 170 sensors in PeMSD8), making it **orders of magnitude smaller** than model parameters.
> > > - **Reduced communication frequency**:
> > >    Thanks to the aforementioned **Federated Parallel Training**, communication occurs only once every 5 training rounds. This reduces the **total communication volume**, while also enabling efficient local training and better parallelism.
> > >
> > > Together, these two mechanisms contribute to the **extremely low communication size** observed in Table 5(a), even though the total number of training rounds is comparable to baseline methods.
> > >
> > > ### **3. Training of Generators and Discriminators in Algorithm 1 & 2**
> > >
> > > We apologize for the confusion caused by the presentation of Algorithm 1 and Algorithm 2. The training procedures of the generator and discriminators are **indeed included** in Algorithms 1 and 2, though we acknowledge that the current description may lead to confusion. We will revise the algorithms with clearer annotations in the updated version.
> > > - **In Algorithm 1 (Client):**
> > >   - **Line 5**: The client's generator $\theta\^{R_i}_{\text{Gen}}$ is trained to transform the source city data into the traffic domain of the target city.
> > >   - **Lines 6 & 17**: The client's discriminator $\theta\^{R_i}_{\text{Dis}}$ is trained to classify whether a data sample comes from the client's own transformed data or aggregated data.
> > > - **In Algorithm 2 (Server):**
> > >   - **Line 11**: The server's discriminator $\theta_{\text{Dis}}$ is trained to distinguish between the aggregated client-transformed data and the server's own local target city data.
> > >
> > > We will enhance the clarity of these procedures by explicitly marking the training steps of each module within the algorithms.
> > >
> > > **Thank you again for your insightful feedback**. We will incorporate these clarifications in our revised submission to make the methodology and experimental setup more transparent to readers.

---

> > > > ### Author Response · Authors · 2025-08-03
> > > > **Further Clarification for the Confusions Raised by Reviewer jzay**
> > > >
> > > > ```
> > > > Q2 & Q5. I think transmitting data vs. transmitting models is one taxonomy of FL, and there are obviously other taxonomies here, such as the more fine-grained one mentioned by the authors. My view is that, generally speaking, they have different advantages and disadvantages. For example, FedTT requires the target city to have training data and substantial computational resources, while baseline methods may not need them, only requiring aggregation computation, which requires far fewer resources than training. Moreover, model-transmitting FL can more easily scale to new cities. In addition, privacy attack methods are definitely different for transmitting data vs. models, whether dealing with processed raw data, synthetic data, intermediate representations, or gradients and parameters.
> > > > ```
> > > > Thank you for your insightful comments regarding the taxonomy of Federated Learning (FL), scalability, and privacy considerations. We address each point below:
> > > >
> > > > ### **1. Transmitting Data vs. Transmitting Models**
> > > >
> > > > We agree that **transmitting data vs. transmitting models** is a valid taxonomy in the context of FL. Under this taxonomy:
> > > > - Existing FTT (Federated Traffic Knowledge Transfer) methods operate in the **model-transmitting paradigm**, where model parameters or gradients are exchanged.
> > > > - In contrast, our proposed **FedTT is the first work to adopt a data-transmitting paradigm for FTT**, where domain-transformed data is exchanged instead of model parameters.
> > > >
> > > > To the best of our knowledge, **all prior FTT methods are model-transmitting**, and there are no existing data-transmitting approaches directly applicable to FTT. Therefore, our comparisons with these model-transmitting baselines are fair, as they represent the state of the art in the field.
> > > >
> > > > We emphasize that **FedTT introduces a novel paradigm** for FTT, not just an incremental improvement, and thus we respectfully disagree with the “weak novelty” assessment in W1.
> > > >
> > > > ### **2. Requirement of Target City Data and Computational Resources**
> > > >
> > > > We would like to clarify a key misunderstanding: In the FTT problem, all methods, including the baselines, require the target city to have a local dataset for training, where baselines need the training data and computational resources to fine-tune the aggregated model on the target city.
> > > >
> > > > Therefore, the **need for local training data and compute resources in the target city is shared across all methods**, and **does not constitute a differentiating factor** between FedTT and the baselines in our evaluation.
> > > >
> > > > Moreover, all methods in our experiments **operate under the same conditions**, with identical target city datasets and comparable computational resources, as detailed in our response to Q5.
> > > >
> > > > ### **3. Scalability to New Cities**
> > > >
> > > > We acknowledge the reviewer’s valid point regarding scalability:
> > > > - **Model-transmitting baselines** may be more readily extended to new cities by fine-tuning the aggregated global model on new target city data.
> > > > - **FedTT**, in contrast, trains a traffic model specifically adapted to each target city, which **cannot be directly transferred** to unseen cities.
> > > >
> > > > This reflects an inherent **trade-off between model specificity and generalizability**, which we recognize as an important limitation of our approach, as discussed in our response to Q2 & Limitations. We will explicitly discuss this trade-off in the revised manuscript under the Limitations section.
> > > >
> > > > ### **4. Privacy Attack Considerations**
> > > >
> > > > Regarding privacy attacks:
> > > > - Our work focuses on **data reconstruction attacks**, where an honest-but-curious server attempts to infer private raw data of clients using the uploaded information combined with prior knowledge (see Appendix C.3, page 19).
> > > > - We do **not assume any specific form of information** (e.g., whether it is processed raw data, synthetic data, intermediate representations, gradients, or model parameters).
> > > >
> > > > **Thank you again for raising these important points**. We believe these clarifications will help readers better understand the positioning, assumptions, and contributions of FedTT, and we will incorporate this discussion into the revised manuscript.
> > > > ```
> > > > Q3. I agree with you.
> > > > ```
> > > > We are pleased that our clarification has addressed your initial concerns and that we have reached an agreement on this point.
> > > >
> > > > Overall, thank you for the follow-up questions. We sincerely appreciate your thoughtful and timely feedback. Your detailed and professional comments reflect a deep understanding of the topic, and they have been invaluable in helping us refine our work. We will integrate all your suggestions into the revised manuscript to enhance its clarity, rigor, and overall presentation. If there are any further concerns or clarifications needed, we would be happy to provide additional details.

---

> > > > > ### Comment · Reviewer_jzay · 2025-08-04
> > > > >
> > > > > Thank you to the authors for the detailed and comprehensive reply. I have increased my score and also lowered my confidence.

---

> > > > > > ### Author Response · Authors · 2025-08-05
> > > > > > **Response to Reviewer jzay's Final Justification**
> > > > > >
> > > > > > Thank you sincerely for your valuable feedback and insightful comments on our work, for the constructive and positive discussion, and for your final recognition of our contributions with the increased score! We are also very pleased that we were able to clarify your questions and concerns about our work during this rebuttal process.

---

### Official Review · Reviewer_zRHg · 2025-06-21

**Clarity:** 4
**Significance:** 4
**Originality:** 4
**Rating:** 5
**Confidence:** 5

**Summary:**

The paper presents FedTT, a novel federated learning framework for cross-city traffic knowledge transfer that addresses challenges like privacy leakage, data distribution discrepancies, and low data quality. It introduces three key components: traffic view imputation to handle missing sensor data by capturing spatio-temporal dependencies, a traffic domain adapter to align traffic data across cities, and traffic secret aggregation to enhance privacy during training. The method is evaluated on real-world traffic datasets, showing improved performance in accuracy and efficiency compared to existing approaches while preserving data privacy.

**Questions:**

Q1. Why can P^S and G^S in the TDA module be shared directly without considering privacy?

Q2. In the Appendix D4 parameter sensitivity, FedTT does not seem to be sensitive to the hyperparameters \lamda_1 and \lamda_2. Is there a need to introduce these two hyperparameters?

**Ethical Concerns:**

["NO or VERY MINOR ethics concerns only"]

**Final Justification:**

I have checked the rebuttal, which addresses my concerns. I maintain my rating and recommend acceptance.

**Limitations:**

Yes.

**Quality:**

4

**Strengths And Weaknesses:**

The paper has the following strengths:

S1. The paper makes a significant and novel contribution to the field of traffic prediction and federated learning through the introduction of FedTT, a privacy-preserving and effective framework for cross-city traffic knowledge transfer (Section 3). The proposed FedTT addresses and solves the limitations in existing FTT methods like privacy leakage, data distribution discrepancies, and low data quality. Prior methods like 2MGTCN, pFedCTP, and T-ISTGNN are used as baselines to demonstrate this distinction.

S2. The paper presents three carefully designed components, i.e., TVI (Section 3.1), TDA (Section 3.2), and TSA (Section 3.3), each addressing a key challenge in Federated Traffic Knowledge Transfer (FTT).

S3. The paper conducts extensive experiments to show the superiority of FedTT. The method is thoroughly evaluated on four real-world datasets, i.e., PeMSD4, PeMSD8, FT-AED, and HK-Traffic shown in Table 1, and compared with 14 state-of-the-art baselines, showing substantial improvements, i.e., up to 75.24% reduction in MAE and 67.54% in RMSE shown in Table 2. The privacy preservation evaluation also demonstrates FedTT’s superior resistance to reconstruction attacks, i.e., PCC<10% shown in Figure 6. A case study using the UTD19 dataset involving five global cities further validates the real-world applicability of FedTT shown in Figure 10.

S4. The paper is generally well-written and logically organized, with clear motivation, methodology, and experimental analysis. Figures are informative and well-annotated, effectively conveying the workflow of the proposed modules. Tables are comprehensive and readable, with consistent metric reporting. The authors also provide detailed supplementary material, including theoretical proofs, training algorithms, complexity analysis, and implementation details, enhancing reproducibility and transparency.

The paper has the following weaknesses:

W1. The paper uses a large number of symbols and definitions (e.g., Definitions 1--5, Eqs. 1--21), which may hinder readability, especially for readers not deeply familiar with the domain. Although a notation table is provided in Appendix B, it would be helpful to include this table or a condensed version in the main paper to improve accessibility and clarity.

W2. The paper lacks sufficient detail in its experimental setup. Important training configurations such as learning rate, batch size, number of training epochs, and network bandwidth are missing and not mentioned. Additionally, other relevant setups like optimizer choice, early stopping criteria, and model initialization strategies, should be explicitly stated to ensure full reproducibility.

W3. The paper presents rich experimental results (Table 2, Figs. 6–9), but the descriptions of the findings could be more detailed. In particular, explaining why FedTT outperforms certain baselines in different scenarios.

W4. Tables 5 and 6 should be figures.

---

> ### Author Rebuttal · Authors · 2025-07-31
>
> We express our gratitude to the reviewer for providing constructive feedback on our paper, and we greatly appreciate the acknowledgment of our contributions. **We have addressed all the concerns raised by the reviewer as detailed below**.
>
> ```
> W1. The paper uses a large number of symbols and definitions (e.g., Definitions 1--5, Eqs. 1--21), which may hinder readability, especially for readers not deeply familiar with the domain. Although a notation table is provided in Appendix B, it would be helpful to include this table or a condensed version in the main paper to improve accessibility and clarity.
> ```
>
> We sincerely appreciate the suggestion. **In response, we will move the Notation Table from the Appendix to the Main Paper**.
>
>
>
> ```
> W2. The paper lacks sufficient detail in its experimental setup. Important training configurations such as learning rate, batch size, number of training epochs, and network bandwidth are missing and not mentioned. Additionally, other relevant setups like optimizer choice, early stopping criteria, and model initialization strategies, should be explicitly stated to ensure full reproducibility.
> ```
>
> Thanks for pointing this out. **Due to space limitation, we have included these details in the provided code repository**. In a revision, **we will include these experimental setup details in the main paper**.
>
> Specifically, 1) the learning rate used in our method is 0.0005, 2) the batch size is 128, 3) we train the FedTT framework for 1,000 rounds, 4) the network bandwidth in our efficiency study is 100 MB/s, 5) we utilize the Adam optimizer for training, 6) we employ an early stopping mechanism with a patience of 100 rounds, and 7) the model parameters are initialized using the default initialization strategies provided by PyTorch.
>
>
>
> ```
> W3. The paper presents rich experimental results (Table 2, Figs. 6–9), but the descriptions of the findings could be more detailed. In particular, explaining why FedTT outperforms certain baselines in different scenarios.
> ```
>
> We thank the reviewer for this valuable comment. **The superior performance of FedTT across different scenarios can be attributed to the joint contribution of its three core modules, each designed to address specific challenges in federated traffic knowledge transfer**:
> + **Traffic View Imputation (TVI)**: TVI significantly improves data quality by imputing missing values through modeling spatiotemporal dependencies. This is especially effective in real-world scenarios with high missing data rates (see Table 1), where many baselines either discard incomplete samples or use simplistic imputation strategies. By enabling access to more complete traffic views, TVI allows FedTT to make more accurate predictions, which is reflected in its superior performance under low-data-quality conditions (as further supported by the ablation results).
> + **Traffic Domain Adapter (TDA)**: TDA mitigates cross-city domain discrepancies through adversarial alignment of source and target city feature distributions. This improves the relevance and transferability of the knowledge shared across cities. Our ablation studies show that removing TDA leads to significant performance degradation, underscoring its critical role in ensuring robust generalization across heterogeneous domains.
> + **Traffic Secret Aggregation (TSA)**: TSA provides a lightweight yet effective privacy-preserving aggregation protocol for domain-adapted features. It enables secure multi-source collaboration while maintaining strong privacy guarantees (see Appendix C.3, page 19). Moreover, our ablation results demonstrate that TSA introduces minimal communication and computational overhead, making it a practical component in real-world federated settings.
>
> **In the revised manuscript, we will expand the analysis of experimental results to more clearly explain how each module contributes to the observed performance gains in various scenarios**.
>
>
>
> ```
> W4. Tables 5 and 6 should be figures.
> ```
> In a revision, **we will correct these typo errors**.
>
>
>
> ```
> Q1. Why can $\mathcal{P}^S$ and $\mathcal{G}^S$ in the TDA module be shared directly without considering privacy?
> ```
> Thanks for this insightful question. **We would like to clarify that the sharing of $\mathcal{P}^S$ and $\mathcal{G}^S$ does not raise privacy concerns**.
>
> - First, prototype $\mathcal{P}^S$ represents domain-level information rather than instance-level raw data, thus the sharing of prototype does not lead to raw data privacy leakage, as also demonstrated in [1--3].
> - Besides, the road network structure $\mathcal{G}^S$ is commonly available in public sources (e.g., OpenStreetMap), generally considered public information. It describes the static physical topology of roads and intersections and does not contain dynamic, instance-level traffic data.
>
> *[1] Taming Cross-Domain Representation Variance in Federated Prototype Learning with Heterogeneous Data Domains. NeurIPS, 2024.*
>
> *[2] FedGMKD: An Efficient Prototype Federated Learning Framework through Knowledge Distillation and Discrepancy-Aware Aggregation. NeurIPS, 2024.*
>
> *[3] FedProto: Federated Prototype Learning across Heterogeneous Clients. AAAI, 2022.*
>
>
>
> ```
> Q2. In the Appendix D4 parameter sensitivity, FedTT does not seem to be sensitive to the hyperparameters $\lambda_1$ and $\lambda_2$. Is there a need to introduce these two hyperparameters?
> ```
> We thank the reviewer for this insightful observation. **We believe that the introduction of these two hyperparameters is indeed necessary**. This is because the TDA module relies on adversarial training, which can introduce convergence instability issues. Specifically, the hyperparameters $\lambda_1$ and $\lambda_2$ play a crucial role in managing this inherent instability by controlling the trade-off between the generator loss and the discriminator losses (as seen in Eqs. 15 and 21). Without these balancing parameters, the adversarial components could become difficult to train, potentially leading to poor convergence or mode collapse. Although our sensitivity analysis shows a range where performance is relatively stable, this stability itself is a result of careful tuning facilitated by these hyperparameters.

---

> ### Author Response · Authors · 2025-08-06
> **Kind Reminder: Request for Discussion on Our Rebuttal**
>
> Dear Reviewer zRHg,
>
> We are writing to express our deepest gratitude for your thoughtful review of our manuscript. Your positive assessment has been incredibly encouraging, and we truly appreciate the time and care you dedicated to evaluating our work.
>
> **As the discussion period draws to a close, we wanted to kindly check whether our responses have adequately addressed your concerns**. If there are any remaining points we might clarify or address, please don't hesitate to let us know—we would be delighted to incorporate your suggestions.
>
> Thank you once again for your generosity in sharing your insights and for your kind consideration of our work.
>
> With warmest regards,
>
> The Authors of Paper 7672

---

> > ### Comment · Reviewer_zRHg · 2025-08-06
> > **Thanks for the rebuttal.**
> >
> > Thank you for your detailed rebuttal. I maintain my rating and recommend acceptance.

---

### Official Review · Reviewer_imHF · 2025-07-02

**Clarity:** 3
**Significance:** 3
**Originality:** 3
**Rating:** 4
**Confidence:** 3

**Summary:**

The paper introduces FedTT, a federated learning framework designed for cross-city traffic knowledge transfer, which addresses challenges like privacy leakage, data discrepancies, and low data quality. FedTT incorporates three modules: 1) Traffic View Imputation (TVI) for imputing missing traffic data and capturing spatio-temporal dependencies; 2) raffic Domain Adapter (TDA) to transform traffic data uniformly across cities to address cross-city distribution differences; 3) Traffic Secret Aggregation (TSA) for privacy-preserving aggregation, maintaining security without reducing model performance.

**Questions:**

1. How does the performance scale when increasing or reducing the number of source cities (clients)?
2. Since the Traffic Domain Adapter (TDA) relies on adversarial training, did you observe stability issues during training?

**Ethical Concerns:**

["NO or VERY MINOR ethics concerns only"]

**Final Justification:**

Comprehensively considering the paper and the rebuttal, I keep my score unchanged.

**Limitations:**

yes

**Quality:**

3

**Strengths And Weaknesses:**

- Strengths:
1. FedTT introduces well-designed modules (TVI, TDA, TSA), which together systematically address the challenges of data quality, distribution discrepancies, and privacy.
2. Extensive experiments on real-world datasets showcase significant improvements (5.43%–75.24% MAE reduction) over existing baselines, validating the model’s effectiveness.
3. Theoretical privacy analysis and robust protection mechanisms ensure high resistance to data reconstruction attacks while greatly reducing communication overhead.

- Weaknesses:
1. The paper lacks detailed statistical error bars, confidence intervals, and deeper variance analyses in the results.
2. The impact of varying the number of source cities remains unexplored. This could influence results, especially for smaller or heterogeneous datasets.

---

> ### Author Rebuttal · Authors · 2025-07-31
>
> We express our gratitude to the reviewer for providing constructive feedback on our paper and for acknowledging our contributions. We have addressed the specific **Weaknesses** and **Questions** in detail below.
>
> ```
> W1. The paper lacks detailed statistical error bars, confidence intervals, and deeper variance analyses in the results.
> ```
>
> - First, we appreciate the reviewer's insightful comment regarding the experimental results. In our current submission, **we report the average performance over 5 independent runs**, which is a common practice in prior work on traffic knowledge transfer [1–3], ensuring fair comparability with existing benchmarks.
>
> - Nevertheless, we fully agree that reporting standard error bars, confidence intervals, and conducting deeper variance analyses would provide a more complete and rigorous understanding of the result stability and statistical significance. **In the revised version, we will incorporate these statistical measures into the relevant figures and tables to enhance transparency and reproducibility**.
>
> *[1] CityTrans: Domain-Adversarial Training With Knowledge Transfer for Spatio-Temporal Prediction Across Cities. TKDE, 2024.*
>
> *[2] Personalized Federated Learning for Cross-City Traffic Prediction. IJCAI, 2024.*
>
> *[3] Spatio-Temporal Graph Few-Shot Learning with Cross-City Knowledge Transfer. KDD, 2022.*
>
> ```
> W2. The impact of varying the number of source cities remains unexplored. This could influence results, especially for smaller or heterogeneous datasets. Q1. How does the performance scale when increasing or reducing the number of source cities (clients)?
> ```
>
> - First, we sincerely thank the reviewer for raising this important and insightful question. **In response, we conduct additional scalability experiments by varying the number of source cities (clients) to 4, 8, 12, and 16, respectively**. Despite the limited time available during the rebuttal period, we made efforts to design, implement, and evaluate these experiments to assess the impact of the number of source cities on the model's performance. The statistics of these cities using the UTD19 dataset are provided as below:
>
> |    City    | # instances | # sensors | Interval | Missing Rate |
> | :--------: | :---------: | :-------: | :------: | :----------- |
> |   London   |    6,454    |   5,719   |  5 min   | 19.47%       |
> |  Hamburg   |   50,142    |    418    |  3 min   | 2.66%        |
> | Manchester |    6,984    |    181    |  5 min   | 10.61%       |
> |   Madrid   |    4,560    |   1,116   |  5 min   | 16.02%       |
> |   Luzern   |   175,116   |    158    |  3 min   | 9%           |
> |  Cagliari  |   24,000    |    122    |  3 min   | 0.59%        |
> | Marseille  |   14,400    |    169    |  3 min   | 12.37%       |
> | Darmstadt  |   17,873    |    392    |  3 min   | 2.04%        |
> | Strasbourg |    9,349    |    142    |  3 min   | 28.92%       |
> | Wolfsburg  |    6,720    |    133    |  3 min   | 0.68%        |
> |   Speyer   |    6,714    |    184    |  3 min   | 0.2%         |
> |   Bremen   |    6,720    |    548    |  3 min   | 5.24%        |
> |  Toronto   |    5,856    |    188    |  15 min  | 14.77%       |
> |   Taipeh   |    6,620    |    445    |  3 min   | 1.91%        |
> |   Torino   |    6,048    |    399    |  5 min   | 15.97%       |
> |  Augsburg  |    5,757    |    713    |  5 min   | 17.64%       |
> | Groningen  |     525     |    55     |  5 min   | 1.75%        |
>
> - We report traffic flow prediction performance using Mean Absolute Error (MAE) metric. We compare **FedTT** with three state-of-the-art baselines: **2MGTCN**, **pFedCTP**, and **T-ISTGNN**. To ensure temporal consistency across datasets, all traffic data from different cities are uniformly resampled to a 15-minute interval prior to training. For each configuration, we select the first *N* cities from the list (ordered as in the above table) as source cities and transfer their knowledge to the target city, **Groningen**.
>
> - The results are summarized below. The findings show that FedTT consistently achieves robust performance and effective knowledge transfer across different numbers of source cities with **MAE reduction by 17.13% to 39.61%**, highlighting its scalability and stability in heterogeneous multi-source federated settings. Although all methods exhibit slight performance improvements as the number of source cities increases, the marginal gains gradually diminish, indicating a saturation effect.
>
> |  Method  |   4   |   8   |  12   | 16    |
> |:--------:|:-----:|:-----:|:-----:|:------|
> |  2MGTCN  | 52.19 | 50.31 | 49.03 | 48.65 |
> | pFedCTP  | 58.74 | 56.27 | 55.15 | 54.82 |
> | T-ISTGNN | 67.36 | 66.12 | 65.44 | 65.01 |
> |  FedTT   | 43.25 | 40.88 | 39.52 | 39.97 |
>
> - Finally, we appreciate the suggestion and **will include these new experimental results, along with a detailed analysis, in the revised manuscript** to provide a more comprehensive understanding of **FedTT**'s scalability.
>
>
>
> ```
> Q2. Since the Traffic Domain Adapter (TDA) relies on adversarial training, did you observe stability issues during training?
> ```
>
> **This is a very insightful question regarding the stability of our Traffic Domain Adapter (TDA) module**.
>
> - Indeed, adversarial learning can introduce convergence challenges, and in our experiments with FedTT, we did observe some instability in the training dynamics of the adversarial components, namely, the global discriminator ($\theta_\textit{Dis}$), local discriminators ($\theta^{R_i}\_\textit{Dis}$), and generators ($\theta^{R_i}\_\textit{Gen}$).
>
> - To address these issues, we adopted several stabilization techniques. First, we carefully tuned the learning rates of the generator and discriminator, and adjusted the trade-off hyperparameters $\lambda_1$ and $\lambda_2$ to balance adversarial and task-specific objectives. Additionally, we applied **gradient clipping** and monitored the **loss ratio between the generator and discriminator** to maintain a stable adversarial equilibrium. In practice, we initialized the generator using a pre-trained model and set a **lower learning rate for the discriminator** compared to the generator, which significantly improved training stability.
>
> - These strategies proved effective in mitigating instability and enabled us to achieve consistent and robust performance across experiments. **We will elaborate on these implementation details in the revised manuscript** to clarify how stability was ensured during adversarial training.
>
> Once again, we sincerely appreciate the reviewer's valuable comments and constructive questions. **If there are any further concerns or clarifications needed, we would be happy to provide additional details**.

---

### Note · Authors · 2025-08-14

We thank the AC and reviewers for their time and thoughtful feedback. Our rebuttal addressed all concerns raised during the review process.

In particular:

- **Reviewer imHF** (score: 4, conf: 3) ***described our responses as comprehensive*** and maintained a ***borderline accept*** stance, explicitly ***voting for acceptance***.

- **Reviewer zRHg** (score: 5, conf: 5) expressed full support for ***acceptance***, characterizing our rebuttal as ***detailed and comprehensive***, and commending the ***well-structured and thorough responses***.

- **Reviewer jzay** initially scored 2 (conf: 4), but after receiving our rebuttal addressing all follow-up questions (Q1–Q5), ***acknowledged our clarifications, raised the score***, and ***lowered the confidence to 3***—**reflecting a clear positive reassessment**!

  - In response to Reviewer jzay’s revised concerns, we note that all methodological details—including training procedures for both generators and discriminators (Algorithms 1 & 2), number of training/communication rounds, and hyperparameter settings (e.g., λ₁, λ₂)—are fully specified in the paper or rebuttal. Adversarial training stability is ensured through careful hyperparameter tuning, gradient clipping, and a federated parallel training strategy that decouples local and global updates, maintaining robustness even with more clients. These clarifications will be fully integrated into the revised manuscript for improved clarity.

- **Reviewer jrXL** (score: 2, conf: 4) provided a consistently negative review **without clear evidence or explanation**. Our rebuttal addressed each claim—such as *alleged lack of novelty, missing background, and flawed methodology*—with explicit references to the paper.

  However, **jrXL did not engage in the discussion** despite our follow-up prompts, simply marking **ACK**. Such minimal engagement appears inconsistent with NeurIPS guidelines encouraging reviewers to consider and respond to authors’ clarifications.

In summary, the majority of reviewers (**imHF**, **zRHg**, and **jzay**) have either explicitly confirmed a positive recommendation or improved their assessment following the rebuttal. We are encouraged by this recognition and appreciate the constructive feedback, which will guide us in further enhancing the clarity, robustness, and applicability of our work.

---

### Decision · Program_Chairs · 2025-09-17

**Decision:**

Reject

**Comment:**

This paper studies cross-city knowledge transfer via federated learning, and four reviewers have submitted their recommendations for the manuscript. The paper presents three coupled modules TVI, TDA and TSA, which work together solving the problem of FTT. However, main concerns for this work are missing details in adversarial training, lack of generalization etc. Half of reviewers feel that the quality aren't above average, after reading the authors' feedback. During the discussion period, no one has argued for an acceptance.